# Modulating HSF1 levels impacts expression of the estrogen receptor *α* and antiestrogen response

Maruhen AD Silveira[1,2] , Christophe Tav[1,2,3,4] , Félix-Antoine Bérube-Simard[1,2], Tania Cuppens[2,3,4], Mickaël Leclercq[2,3,4], Éric Fournier[1,2,3], Maxime C Côté[1,2], Arnaud Droit[2,3,4,5] , Steve Bilodeau[1,2,4,6]

**Master transcription factors control the transcriptional program and are essential to maintain cellular functions. Among them, steroid nuclear receptors, such as the estrogen receptor *α* (ER*α*), are central to the etiology of hormone-dependent cancers which are accordingly treated with corresponding endocrine therapies. However, resistance invariably arises. Here, we show that high levels of the stress response master regulator, the heat shock factor 1 (HSF1), are associated with antiestrogen resistance in breast cancer cells. Indeed, overexpression of HSF1 leads to ER*α* degradation, decreased expression of ER*α*-activated genes, and antiestrogen resistance. Furthermore, we demonstrate that reducing HSF1 levels reinstates expression of the ER*α* and restores response to antiestrogens. Last, our results establish a proof of concept that inhibition of HSF1, in combination with antiestrogens, is a valid strategy to tackle resistant breast cancers. Taken together, we are proposing a mechanism where high HSF1 levels interfere with the ER*α*-dependent transcriptional program leading to endocrine resistance in breast cancer.**

## Introduction

Transcription factors are powerful modulators of the gene expression program which are commonly misregulated in human cancers (Lee & Young, 2013; Bradner et al, 2017; Silveira & Bilodeau, 2018; Lambert et al, 2018). Indeed, cancer cells are often addicted to specific transcription factors to support an uncontrolled growth, making them highly sensitive to their loss-of-function (Bradner et al, 2017). Among transcription factors, master transcription factors have the ability to impose a transcriptional program and to reprogram a cell state (Graf & Enver, 2009; Lee & Young, 2013; Lambert et al, 2018). Classical examples include reprogramming of fibroblasts toward muscle or induced pluripotent cells using a unique (MYOD) (Davis et al, 1987) or limited set (OCT4, SOX2, MYC,

and KLF4) (Takahashi & Yamanaka, 2006) of transcription factor(s). During reprogramming, master regulators not only activate a new program, but also repress the gene signatures associated with the cell-of-origin leading to a stable cell state (Voss & Hager, 2014; Stadhouders et al, 2019). Therefore, a single master transcription factor has the dual ability to activate new and decommission old gene expression programs.

The estrogen receptor *α* (ER*α*) is a ligand-induced master transcription factor controlling breast cancer growth (Siersbæk et al, 2018). Accordingly, antiestrogen therapies using selective estrogen receptor modulators (SERM), selective estrogen receptor degraders, or downregulators (SERD) and aromatase inhibitors limit the action of estrogens (Musgrove & Sutherland, 2009; Osborne & Schiff, 2011; Hanker et al, 2020). However, these drugs only have temporary effects as resistance invariably arise (Musgrove & Sutherland, 2009; Osborne & Schiff, 2011; Hanker et al, 2020). Interestingly, one third of endocrine resistant breast cancers are ER*α*-positive (Encarnación et al, 1993; Johnston et al, 1995), raising questions about the mechanisms which allow them to keep proliferating despite interfering with the estrogen response. The answer is complex as acquiring endocrine resistance is a multistep process involving different molecular mechanisms (Hanker et al, 2020). For example, mutations in the ligand-binding domain of ER*α* (Robinson et al, 2013; Toy et al, 2013) and translocation implicating *ESR1* (Li et al, 2013; Veeraraghavan et al, 2014) are found in a small proportion of ER*α*-positive endocrine-resistant cancers. Other mechanisms include increased expression of cytokines (e.g., TNF*α* and IL-1*β*), growth factors (e.g., EGF, IGF1, and TGF*β*), and their cognate receptors (e.g., EGFR, IGF1-R, HER2, and FGFR) leading to ligand-independent activation of ER*α* or modulation of alternative pathways (Musgrove & Sutherland, 2009; Siersbæk et al, 2018). Furthermore, activation or overexpression of transcription factors (e.g., STAT, NF*k*B, HIF1, and FOXA1) was shown to control ER*α*-regulated genes (Siersbæk et al, 2018). Last, amplification or overexpression of cell-cycle regulators (e.g., MYC, Cyclin D1, Cyclin E1 and Cyclin-Dependent Kinases) are also associated with endocrine-resistant breast

[1]Centre de Recherche du CHU de Québec – Université Laval, Axe Oncologie, Québec, Canada    [2]Centre de Recherche sur le Cancer de l'Université Laval, Québec, Canada    [3]Centre de Recherche du CHU de Québec – Université Laval, Axe Endocrinologie et Néphrologie, Québec, Canada    [4]Centre de Recherche en Données Massives de l'Université Laval, Québec, Canada    [5]Département de Médecine Moléculaire, Faculté de Médecine, Université Laval, Québec, Canada    [6]Département de Biologie Moléculaire, Biochimie Médicale et Pathologie, Faculté de Médecine, Université Laval, Québec, Canada

Correspondence: steve.bilodeau@crchudequebec.ulaval.ca

cancers (Musgrove & Sutherland, 2009). Although the molecular details of these mechanisms differ and are context-dependent, they often share a common theme: the control of the transcriptional program is central to breast cancer etiology and treatment. However, whether additional transcriptional regulators are involved in the endocrine-resistance phenotype of ERα-positive breast cancers remains to be determined.

The master regulator of the stress response heat shock factor 1 (HSF1) is overexpressed in multiple cancers, controlling the transcription of genes involved in cell proliferation, survival and energy metabolism (Dai et al, 2007; Li et al, 2017). Deletion of HSF1 suppresses tumor development in the digestive system, blood, skin, pancreas, and breast (Dai et al, 2007; Dong et al, 2019). Interestingly, HSF1 overexpression is associated with resistance to various treatments and poor prognosis in most of these cancers (Dong et al, 2019). However, the molecular mechanisms underlying the resistance to these therapies often remain elusive.

The association between overexpression of HSF1 and poor prognosis in breast cancer is linked with Lapatinib (HER2 and EGFR inhibitor) (Yallowitz et al, 2018) and Trastuzumab (HER2 antibody) (Zhao et al, 2011) resistance. In addition, high levels of HSF1 in ERα-positive breast cancers are associated with poor prognosis for patients treated with tamoxifen (Santagata et al, 2011; Mendillo et al, 2012; Gökmen-Polar & Badve, 2016). Mechanistically, HSF1 is important for maintaining stem cell–like properties, tumor growth, and metastasis in breast cancer (Santagata et al, 2011; Mendillo et al, 2012; Wang et al, 2015; Gökmen-Polar & Badve, 2016). Accordingly, numerous genes regulated by HSF1, including *HSPD1*, *HSPH1*, and *HSP90AA1*, are also associated with poor prognosis for breast cancer patients (Mendillo et al, 2012; Zoppino et al, 2018). Therefore, evidence suggests that HSF1 is an important transcription factor in the progression toward aggressive forms of breast cancer including endocrine resistance.

Here we show that HSF1 overexpression was sufficient to degrade ERα and initiate antiestrogen resistance in breast cancer cells. In addition, HSF1 was overexpressed in a model of endocrine-resistant breast cancer cells, which correlated with a decrease in ERα-activated genes. Interestingly, depletion of HSF1 in endocrine-resistant cells restored expression of the ERα and, accordingly, reinstated the antiproliferative effects of antiestrogens. Furthermore, our results established that inhibition of HSF1 re-sensitized resistant cells to antiestrogens. Taken together, our results support a model where overexpression of HSF1 inhibits the ERα transcriptional program leading to endocrine resistance in a process that is reversible.

# Results

## HSF1 overexpression is sufficient to trigger partial antiestrogen resistance

Because HSF1 overexpression is associated with poor prognosis and shorter life expectancy for breast cancer patients (Santagata et al, 2011; Mendillo et al, 2012; Gökmen-Polar & Badve, 2016), we hypothesized that it was involved in the molecular etiology leading to antiestrogen resistance. To directly test the hypothesis, we

overexpressed HSF1 in the breast adenocarcinoma MCF7 cells (Fig 1A). High levels of HSF1 were achieved and noticeably correlated with a 40% decrease in ERα protein levels. Accordingly, well-characterized HSF1 (*HSPH1*, *HSP90AA1*, and *HSPD1*) and ERα (*TFF1*, *GREB1*, and *ESR1*) activated genes were, respectively, up- and down-regulated at the RNA level (Fig 1B). On the other hand, genes known to be repressed by ERα (*CCNG2*, *MMD*, and *SMAD6*) were up-regulated, suggesting a release from repression. Overexpression of HSF1 slightly increased proliferation of MCF7 cells when compared with control (doubling time of 37.3 compared with 42.4 h) (Fig 1C and D). Following tamoxifen (a SERM) and fulvestrant (a SERD) treatments, the proliferation of control MCF7 cells was halted, with no significant increase in cell counts after 5 d. However, in HSF1-overexpressing cells, the antiestrogens had limited effects as proliferation was maintained. HSF1 has been shown to be important for the ability of MCF7 cells to grow independently from a solid surface in colony formation soft-agar assays (Whitesell et al, 2014). To corroborate our findings, we conducted soft agar assays for MCF7 cells overexpressing HSF1 combined with tamoxifen and fulvestrant treatments (Fig 1E and F). HSF1 overexpression was associated to an ~36% increase in the number of colonies. Treatment with both tamoxifen and fulvestrant reduced colony formation in control cells by at least 83%. However, when HSF1 was overexpressed, 2.3- and 5.6-fold increases were observed in the number of colonies for tamoxifen- and fulvestrant-treated cells, respectively. To confirm these observations, we overexpressed HSF1 in T47D cells, a model of breast ductal carcinoma. Similar results were observed as ERα expression was decreased (Fig S1A) and cells became partially resistant to fulvestrant (Fig S1B and C). These results support that HSF1 overexpression is sufficient to instate partial resistance to antiestrogens.

## High HSF1 levels destabilize the ERα protein

Because our HSF1 overexpression model represented a cellular adaptation over the course of multiple days, we reasoned that an acute HSF1 activation would be a suitable option to investigate the dynamic relationship with the ERα. Short heat shock pulses have been previously used to activate HSF1 and study the transcriptional response (Mahat et al, 2016). MCF7 cells were incubated at 42°C for up to 60 min, and protein levels were monitored by Western blot (Fig 2A). Within the first 5 min, HSF1 was induced as previously described (Petesch & Lis, 2008; Mahat et al, 2016). After 20 min, ERα levels started to decrease to reach 30% at 60 min, suggesting gene repression, protein degradation, or both. Interestingly, HSF1 has been reported to be able to directly repress transcription by recruiting MTA1 (Khaleque et al, 2008). To determine if induction of HSF1 first led to a decrease in transcription of *ESR1* prior to a decrease of the ERα protein levels, we measured nascent RNA levels (Fig 2B). At 30 min after heat shock, ERα protein levels were reduced by about 40% (Fig 2A), but the nascent transcript for *ESR1*, *TFF1*, and *GREB1* were unchanged (Fig 2B). At the same time, HSF1-activated genes were up-regulated. However, after 60 min of heat shock, nascent RNA levels of ERα-activated genes were decreased (Fig 2B), suggesting that ERα protein levels were affected before transcriptional repression. If HSF1 was directly repressing ERα-activated genes, we would expect HSF1 recruitment at the regulatory region. Noticeably, we did not detect significant levels of HSF1 at the regulatory region of ERα-activated genes following heat

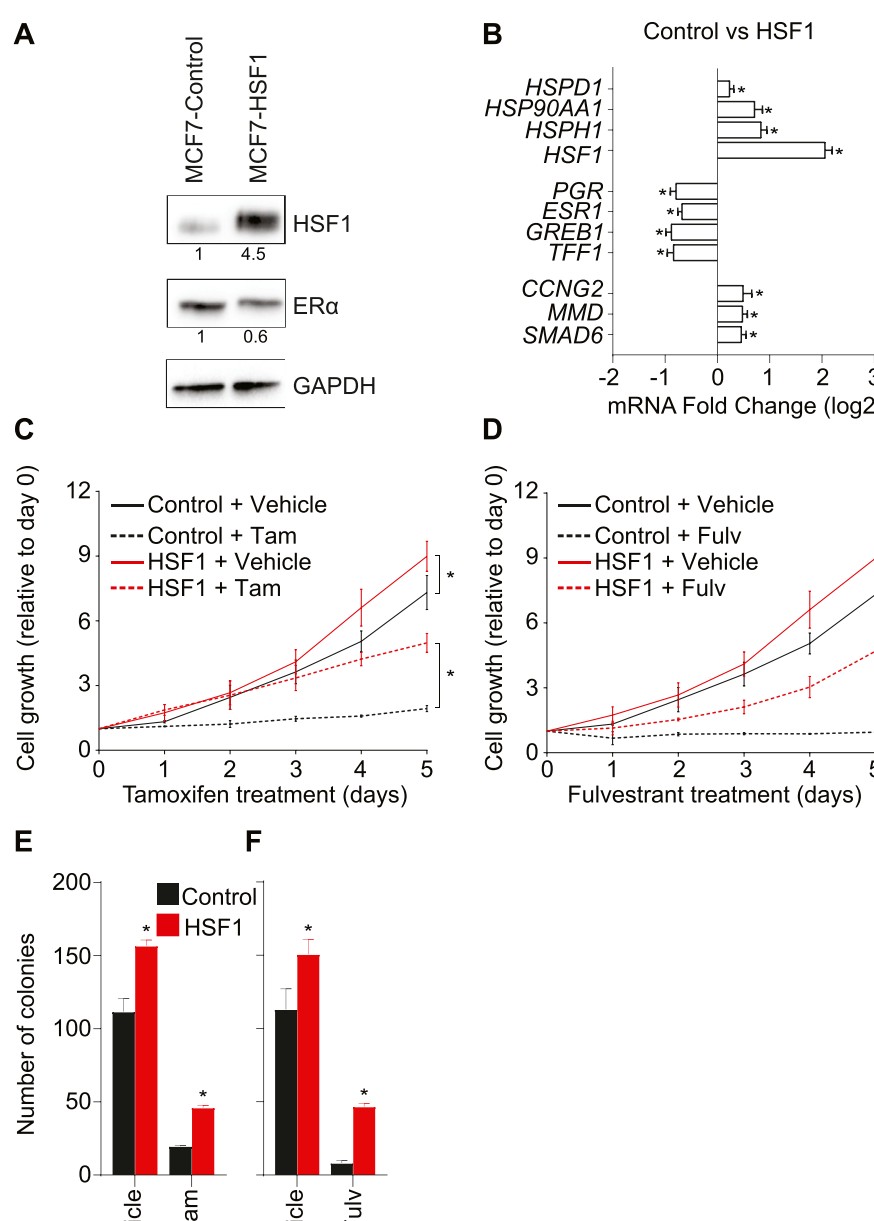

**Figure 1. Heat shock factor 1 (HSF1) overexpression induces resistance to antiestrogens in MCF7 cells.**
**(A)** Western blot analysis of total protein extracts showing that HSF1 is overexpressed 4.5-fold in MCF7 cells overexpressing HSF1 (MCF7-HSF1) compared with control cells (MCF7-control). In addition, a 40% decrease in estrogen receptor α (ERα) is observed compared with control cells. Protein quantifications were performed using the ImageJ software and GAPDH was used as a loading control. **(B)** HSF1-activated genes (*HSF1*, *HSPH1*, *HSP90AA1*, and *HSPD1*) are up-regulated in MCF7 cells overexpressing HSF1, whereas ERα-activated genes (*ESR1*, *TFF1*, *GREB1*, and *PGR*) are down-regulated. This is in contrast with ERα-repressed genes (*CCNG2*, *MMD*, and *SMAD6*) which are up-regulated. Total mRNA levels were measured using qRT-PCR. Data represent the $\log_2$ fold change between MCF7-control and MCF7-HSF1. Means ± SEM (n = 3–7 biological replicates) are represented and *P*-values were calculated using a bivariate *t* test. **(C, D)** HSF1-overexpressing MCF7 cells are resistant to a high dose (1,000 nM) of (C) tamoxifen (Tam) and (D) fulvestrant (Fulv). Cells were counted manually each day and values are represented as a fold compared with Day 0. Means ± SD (n = 3 biological replicates) are represented and *P*-values were calculated using univariate *t* test. **(E, F)** Overexpression of HSF1 increases MCF7 cells colony formation in soft-agar assays after antiestrogen treatments. **(E, F)** MCF7-control and MCF7-HSF1 cells were treated with 1,000 nM of (E) tamoxifen or (F) fulvestrant for 15 d before colony counting. Means ± SD (n = 3 biological replicates) are represented and *P*-values were calculated using a univariate *t* test. (*) *P* ≤ 0.05.
Source data are available for this figure.

shock (Fig 2C). Indeed, the heat shock led to a massive increase in HSF1 by ChIP-qPCR at known target genes within 15 min, whereas close to background signal was observed at ERα-activated genes. It is to be noted that a heat shock is known to induce multiple pathways and transcription factors in addition to HSF1 (Mahat et al, 2016; Gomez-Pastor et al, 2018); one of which could be regulating ERα-activated genes. As expected, the heat shock decreased recruitment of ERα at the ERα-activated genes (Fig 2D). To confirm that HSF1 was sufficient to decrease recruitment of ERα at ERα-activated genes, we overexpressed HSF1 in MCF7 cells. Similarly, overexpression of HSF1 in MCF7 cells led to increased recruitment at known targets (Fig 2E). In support of the heat shock results, overexpression of HSF1 led to a decrease recruitment of ERα

at ERα-activated genes (Fig 2F). These results suggest that activation of HSF1 decreases recruitment of ERα at regulatory regions leading to transcriptional changes.

The ubiquitin–proteasome pathway is known to regulate transcriptional control at multiple steps (Geng et al, 2012). To support the model that the ERα protein levels are destabilized by HSF1, MCF7 cells overexpressing HSF1 and their matching control were treated with the proteasome inhibitor MG-132 (Fig 2G). After a 4-h MG-132 treatment, the ERα levels were increased by 40% supporting its constant turnover in MCF7 cells. As expected, overexpression of HSF1 led to a 50% decrease in the ERα protein levels. However, when HSF1-overexpressing cells were treated with MG-132, the ERα protein levels recovered to a level similar to control cells suggesting the implication of the proteasome in the process.

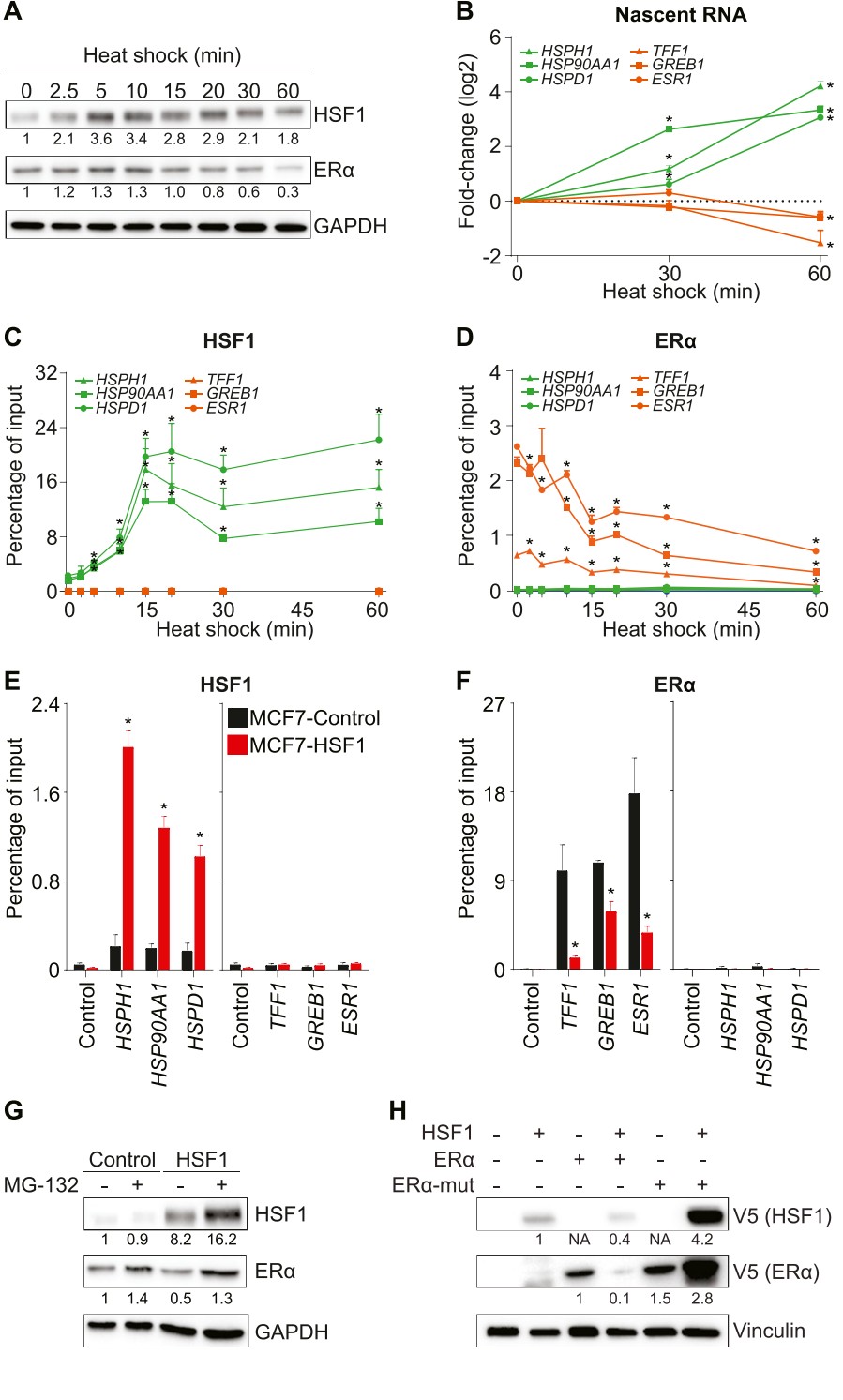

**Figure 2. Increased levels of heat shock factor 1 (HSF1) leads to estrogen receptor α (ERα) degradation.**

**(A)** HSF1 protein levels are quickly increased, whereas ERα protein levels are reduced after heat shock in MCF7 cells. Western blots of total protein extracts were quantified using the ImageJ software and GAPDH as a loading control. **(B)** Quantification of nascent RNA transcripts after heat shock. Transcription of HSF1-activated genes (*HSPH1*, *HSP90AA1*, and *HSPD1*) is up-regulated at 30 and 60 min. However, transcription of ERα-activated genes (*ESR1*, *TFF1* and *GREB1*) is decreased only 60 min after heat shock. Nascent mRNA levels were measured using qRT-PCR. Data represent the log2 fold change in MCF7 cells. Means ± SEM (n = 3 biological replicates) are represented and *P*-values were calculated using a bivariate *t* test. **(C, D, E, F)** HSF1 is recruited to HSF1-activated genes, but not to the regulatory region of ERα-activated genes after heat shock. **(D)** ERα levels are decreased at ERα-activated genes after heat shock. **(E, F)** HSF1 is recruited to HSF1-activated genes, whereas (F) ERα is depleted from ERα-activated genes in MCF7 cells overexpressing HSF1. For (C, D, E, F), HSF1 and ERα recruitment were measured by ChIP-qPCR at these ERα-controlled noncoding regulatory regions: the promoter of *TFF1*, the enhancer of *GREB1*, and the enhancer of *ESR1* (Fournier et al, 2016) in addition to the promoters of HSF1-activated genes (*HSPH1*, *HSP90AA1*, and *HSPD1*), which were all described to be occupied and regulated by HSF1 (Mahat et al, 2016). Percentage of input means ± SEM (n = 3–5 biological replicates) are represented and *P*-values were calculated using a bivariate *t* test. **(G)** ERα protein levels are increased by a 4-h treatment with the proteasome inhibitor MG-132 (10 μM) in MCF7-control and MCF7-HSF1 cells. Protein quantifications were performed using the ImageJ software and GAPDH was used as a loading control. **(H)** Western blot analysis showing that HSF1 controls ERα protein stability and turnover. Decreased levels of wild-type ERα are observed when HSF1 is overexpressed. However, levels of ERα mutated on lysines 302 and 303 (ERα-mut) are not reduced. Protein quantifications were performed using the ImageJ software and vinculin was used as a loading control (NA, no available quantification). *P ≤ 0.05.

To confirm that HSF1 induces degradation of the ERα protein, we mutated the lysines 302 and 303 of ERα which were shown to promote stability and regulate turnover (Berry et al, 2008) (Fig 2H). Although overexpression of HSF1 decreased the level of the wild-type ERα protein, the mutant was not reduced. Therefore, our results suggest that high levels of HSF1 triggers degradation of ERα through the proteasome.

## Acquisition of antiestrogen resistance is associated with increased HSF1 and decreased ERα levels

Whether or not an increase in HSF1 is a central mechanism to acquire and maintain endocrine resistance is an open question. We selected the LCC cellular models (Brünner et al, 1993, 1997) of endocrine resistance to determine the importance of HSF1. These

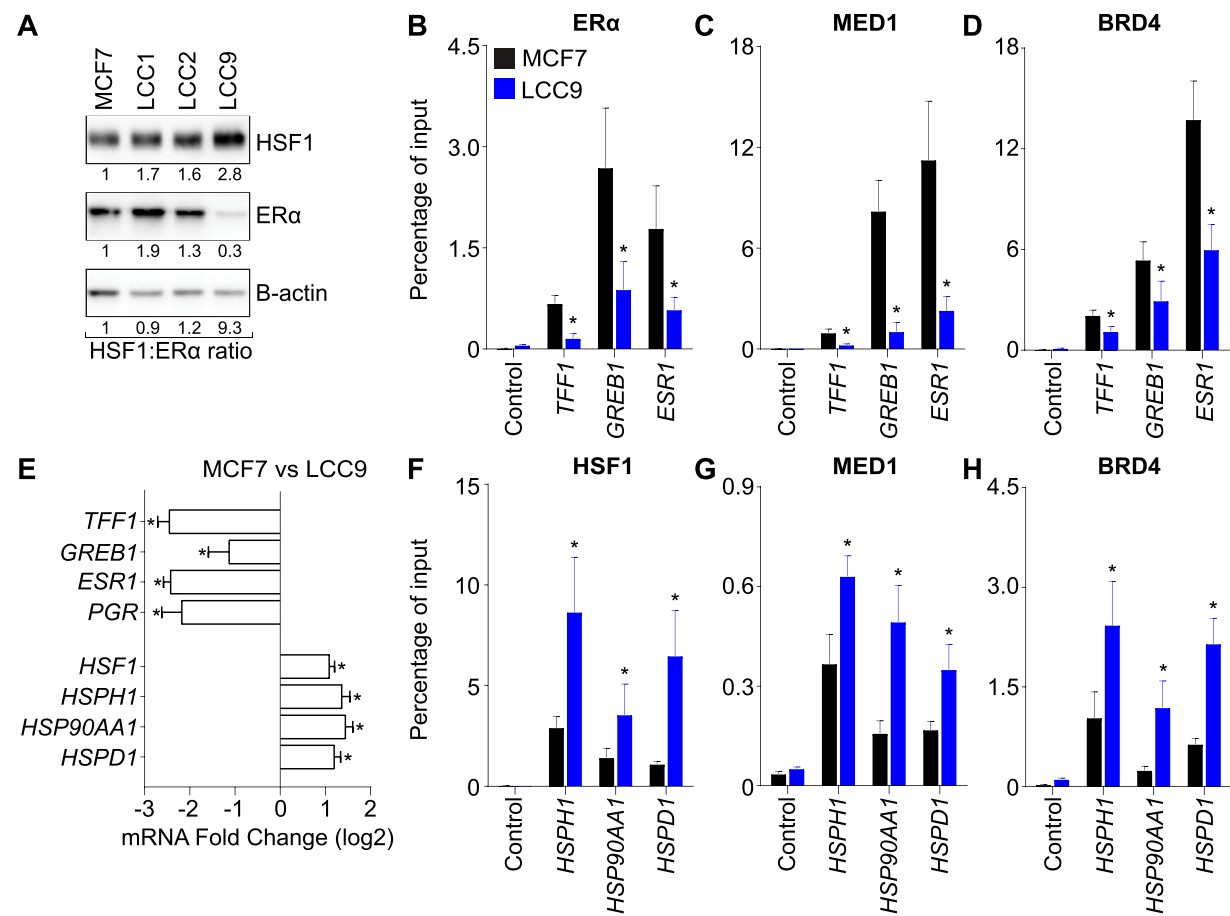

**Figure 3. Activation of the heat shock factor 1 (HSF1) pathway is associated with endocrine-resistance in breast cancer LCC9 cells.**
**(A)** HSF1 protein levels are increased, whereas estrogen receptor α (ERα) protein levels are reduced in antiestrogen-resistant LCC9 cells. Western blots of total protein extracts were quantified using the ImageJ software and β-actin as loading control. The HSF1:ERα protein ratio is indicated. **(B, C, D)** Recruitment of (B) ERα, (C) MED1, and (D) BRD4 are decreased at well-characterized ERα-activated genes in LCC9 compared with MCF7 cells. Recruitment was measured by ChIP-qPCR at the following ERα-controlled noncoding regulatory regions: the promoter of *TFF1*, the enhancer of *GREB1*, and the enhancer of *ESR1* (Fournier et al, 2016). Percentage of input means ± the SEM (n = 4–6 biological replicates) are represented and *P*-values were calculated using a bivariate *t* test. **(E)** ERα-activated genes (*ESR1*, *TFF1*, *GREB1*, and *PGR*) are down-regulated and HSF1-activated genes (*HSF1*, *HSPH1*, *HSP90AA1*, and *HSPD1*) are up-regulated in LCC9 compared with MCF7 cells. Total mRNA levels were measured using qRT-PCR. Data represent the $\log_2$ fold change between MCF7 and LCC9 cells. Means ± SEM (n = 5–6 biological replicates) are represented and *P*-values were calculated using a bivariate *t* test. **(F, G, H)** Recruitment of (F) HSF1, (G) MED1, and (H) BRD4 are increased at HSF1-activated genes in LCC9 compared with MCF7 cells. Recruitment was measured by ChIP-qPCR using the promoters of *HSPH1*, *HSP90AA1*, and *HSPD1* which were all described to be occupied and regulated by HSF1 (Mahat et al, 2016). Percentage of input means ± SEM (n = 3–6 biological replicates) are represented and *P*-values were calculated using a bivariate *t* test. *$P ≤ 0.05$.

cells were derived from MCF7 cells using selective pressure to create estrogen-independent and estrogen-responsive (LCC1), tamoxifen-resistant (LCC2), and tamoxifen and fulvestrant cross-resistant (LCC9) cells. To validate that the MCF7/LCC models represented the evolution observed in patients, we measured the HSF1 and ERα protein levels. HSF1 gradually increased with the endocrine resistance phenotypes, whereas the ERα was reduced (Fig 3A). These changes corresponded to a 9.3-fold increase in the ratio between HSF1 and ERα in LCC9 compared with MCF7 cells. These results are in accordance with the association between HSF1-positive tumors and an increased mortality for patients with ERα-positive breast cancers under hormonal therapy (Santagata et al, 2011). Therefore, the MCF7/LCC9 models recapitulate clinical observations made for HSF1 and ERα in breast cancer.

To determine if high HSF1 levels were associated with changes in gene expression during the transition between MCF7 and LCC9 cells, we compared the same set of well-characterized ERα (*TFF1*, *GREB1*, and *ESR1*)- and HSF1 (*HSPH1*, *HSP90AA1*, and *HSPD1*)-activated genes for recruitment of the associated transcription factor, transcriptional coregulators, and total RNA levels. A strong decrease in ERα binding was observed between MCF7 and LCC9 at ERα-activated genes (Fig 3B). This decrease was associated with reduced levels of the transcriptional coregulators MED1 (Fig 3C), a subunit of the mediator complex, and the bromodomain-containing protein 4 (BRD4) (Fig 3D). In addition, the decrease in transcriptional regulators was correlated with lower levels of total RNA in LCC9 cells (Fig 3E). For HSF1-activated genes, opposite results were observed. Indeed, HSF1 recruitment was increased in LCC9 compared with MCF7 cells (Fig 3F). The increase in transcription factors binding was associated with recruitment of MED1 and BRD4 (Fig 3G and H) and higher RNA levels (Fig 3E). These results suggest that the transcriptional program shifted toward HSF1 at the expanse of ERα when an antiestrogen selective pressure was applied on MCF7 cells. To

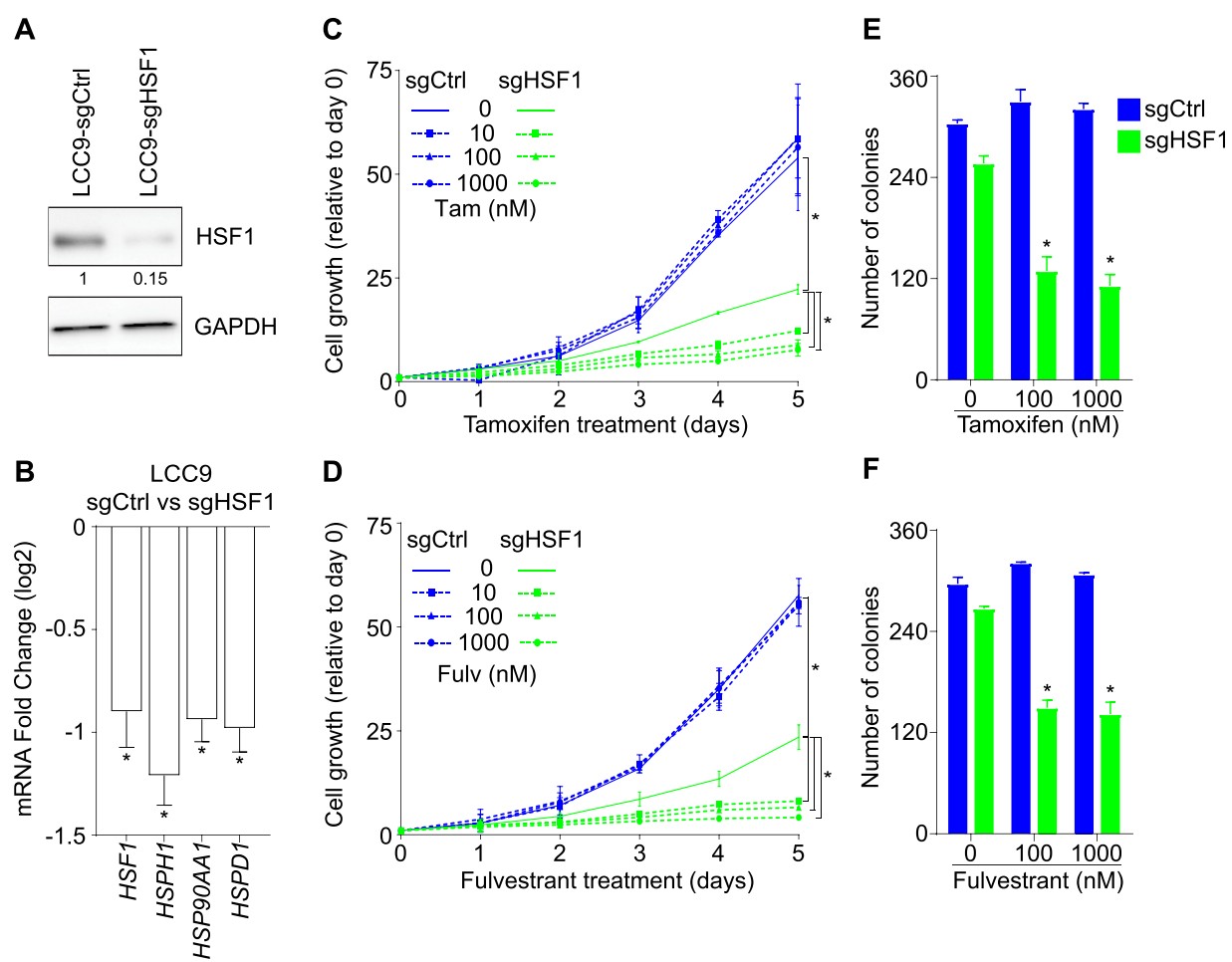

**Figure 4. Heat shock factor 1 (HSF1) is essential to maintain the endocrine resistance phenotype in LCC9 cells.**
**(A)** Western blot analysis of total protein extracts showing an 85% loss of HSF1 protein levels when LCC9 cells are transduced with a single guide RNA targeting HSF1 (sgHSF1) compared with a single guide control (sgCtrl). Protein quantifications were performed using the ImageJ software and GAPDH as a loading control. **(B)** HSF1-activated genes are down-regulated in HSF1-depleted LCC9 cells. Total mRNA levels were measured using qRT-PCR. Data represent the log$_2$ fold change between LCC9 sgCtrl and sgHSF1 cells. Means ± SEM (n = 7 biological replicates) are represented and *P*-values were calculated using a bivariate *t* test. **(C, D)** Restoration of the response to antiestrogens in HSF1-depleted LCC9 cells. **(C)** tamoxifen (Tam) and (D) fulvestrant (Fulv) dose–response curves (10, 100, and 1,000 nM) in LCC9 sgCtrl and sgHSF1 cells. Cells were counted manually each day and values are represented as a fold compared with Day 0. Means ± SD (n = 2–3 biological replicates) are represented and *P*-values were calculated using a univariate *t* test. **(E, F)** Depletion of HSF1 reduces LCC9 cells colony formation in soft-agar assays after antiestrogen treatments. **(E, F)** LCC9 sgCtrl and sgHSF1 were treated with 100 and 1,000 nM of (E) tamoxifen or (F) fulvestrant for 15 d before colony counting. Means ± SD (n = 2 biological replicates) are represented and *P*-values were calculated using a univariate *t* test. \**P* ≤ 0.05.

validate the conclusion, we surveyed gene expression changes in ERα-positive breast cancers (Fig S2A and B). We stratified samples based on HSF1 levels and measured the correlation with known HSF1- and ERα-regulated genes (Abba et al, 2005; Mendillo et al, 2012). A similar trend in gene expression changes was observed: high HSF1 levels were associated with significant increases in HSF1-regulated genes and decreases ERα-regulated genes (Fig S2). Taken together, these results establish that progression toward an endocrine resistance phenotype in breast cancer is associated with an increase in HSF1 and a decrease in ERα transcriptional activity.

### Deletion of HSF1 restores ERα dependency

To establish if HSF1 was responsible for the endocrine resistance phenotype observed in LCC9 cells, we created a loss-of-function

(Fig 4A and B). Depletion of HSF1 (Fig 4A) decreased RNA levels of known HSF1-activated genes (Fig 4B). In addition, proliferation of HSF1-depleted LCC9 (doubling time 24.7 h) was reduced compared with control cells (doubling time 19.2 h) (Fig 4C and D). To determine if the loss of HSF1 was associated with a reversal of the endocrine resistance phenotype, cells were treated with increased concentrations of tamoxifen and fulvestrant for 5 d (Fig 4C and D). Interestingly, loss of HSF1 restored sensitivity to antiestrogens. Indeed, proliferation of LCC9 cells depleted in HSF1 was further reduced (ranging from 45 to 81% compared with control cells) when treated with increasing concentrations of tamoxifen and fulvestrant. To corroborate our findings, we conducted soft agar assays for LCC9 cells depleted of HSF1 combined with tamoxifen and fulvestrant treatments (Fig 4E and F). Colony counts for LCC9 cells depleted in HSF1 were lower than control cells for both tamoxifen

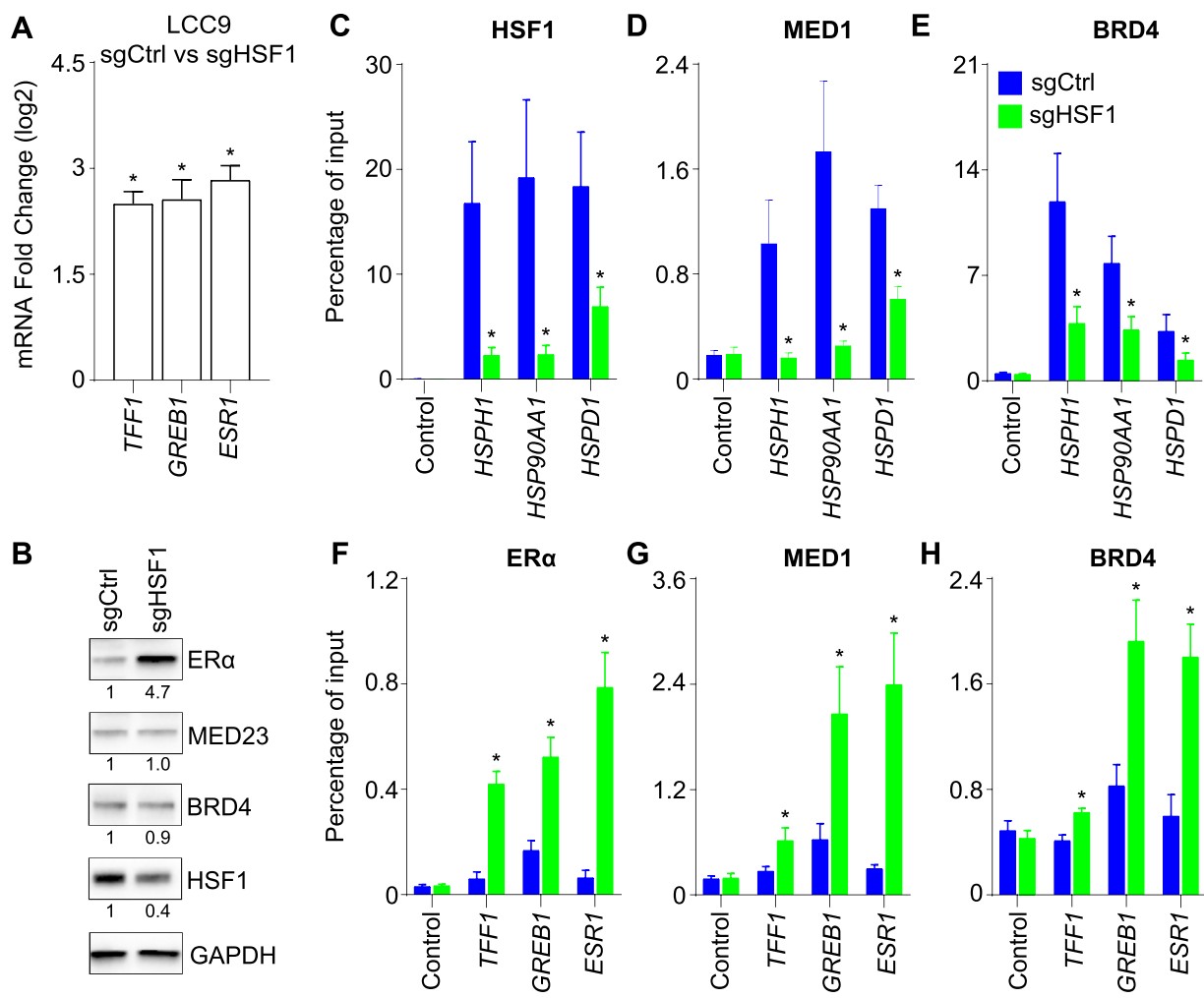

**Figure 5. Depletion of heat shock factor 1 (HSF1) increases estrogen receptor α (ERα) levels and transcriptional activation of ERα-regulated genes.**
**(A)** ERα-activated genes are increased after HSF1 depletion in LCC9 cells. Total mRNA levels were measured using qRT-PCR. Data represent the log$_2$ fold change between LCC9-sgCtrl and sgHSF1. Means ± SEM (n = 5–7 biological replicates) are presented and *P*-values were calculated using a bivariate *t* test. **(B)** Increased expression of ERα in HSF1-depleted LCC9 cells. Western blot analysis of total protein extracts showing a 4.7-fold increase in the ERα expression, whereas transcriptional coregulators BRD4 (0.9-fold) and MED23 (1.0-fold) are not affected when comparing LCC9-sgCtrl and sgHSF1 cells. Protein quantifications were performed using the ImageJ software and GAPDH as a loading control. **(C, D, E)** Recruitment of (C) HSF1, (D) MED1, and (E) BRD4 are decreased at HSF1-activated genes in LCC9-sgHSF1 compared with sgCtrl. Recruitment was measured by ChIP-qPCR using the promoters of *HSPH1*, *HSP90AA1*, and *HSPD1* which were all described to be occupied and regulated by HSF1 (Mahat et al, 2016). Percentage of input means ± SEM (n = 4–7 biological replicates) are represented and *P*-values were calculated using a bivariate *t* test. **(F, G, H)** Recruitment of (F) ERα, (G) MED1, and (H) BRD4 are increased at ERα-activated genes in LCC9-sgHSF1 compared with sgCtrl. Recruitment was measured by ChIP-qPCR at ERα-controlled noncoding regulatory regions: the promoter of *TFF1*, the enhancer of *GREB1*, and the enhancer of *ESR1* (Fournier et al, 2016). Percentage of input means ± SEM (n = 3–7 biological replicates) are represented and *P*-values were calculated using a bivariate *t* test. **P* ≤ 0.05.

and fulvestrant treatments averaging a 58% loss. The ability to restore antiestrogen response by depleting HSF1 suggests that the increased expression of HSF1 in LCC9 cells is sufficient to control proliferation independently from ERα.

**Reinstating the ERα-dependent gene regulation**

Because the LCC9 cells depleted in HSF1 regained sensitivity to antiestrogens, we hypothesized that the ERα-dependent transcriptional program was restored. Gene expression analysis confirmed that total RNA transcripts of ERα-activated genes were increased (Fig 5A) in contrast with HSF1-activated genes (Fig 4B). Loss of HSF1 in LCC9 cells was associated with increased RNA and

protein levels of the ERα (Fig 5A and B) supporting the increase levels of ERα-activated genes (Fig 5A). To determine the molecular consequences of modifying the HSF1:ERα ratios on the transcriptional program of LCC9 cells, we measured recruitment of HSF1, ERα, and their associated coregulators at target genes in HSF1-depleted LCC9 cells. As expected, for HSF1-activated genes, we observed a depletion in HSF1, MED1, and BRD4 (Fig 5C and E). For ERα-activated genes, ERα binding was significantly increased and associated with recruitment of MED1 and BRD4 (Fig 5F and H). Levels of transcriptional coregulators were not affected by the loss of HSF1 (Fig 5B). These results strongly suggest that the transcriptional regulation of ERα-activated genes was restored. To validate that the gain in ERα-dependent transcriptional regulation in absence of HSF1

**A**

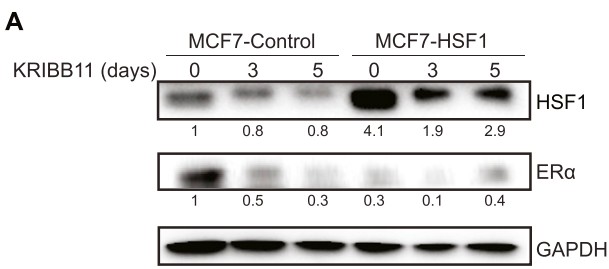

**B**

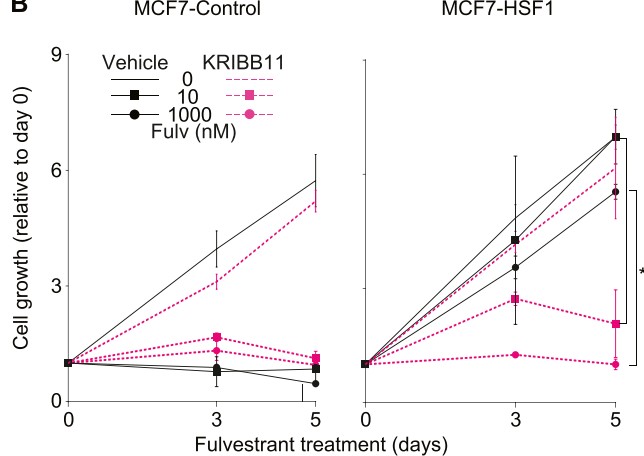

**C**

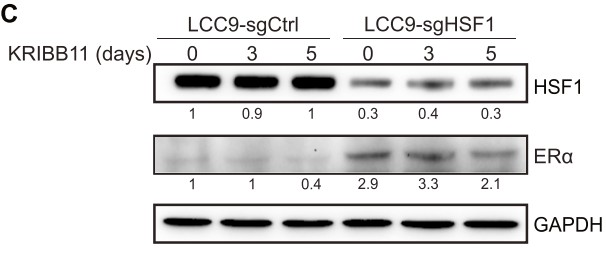

**D**

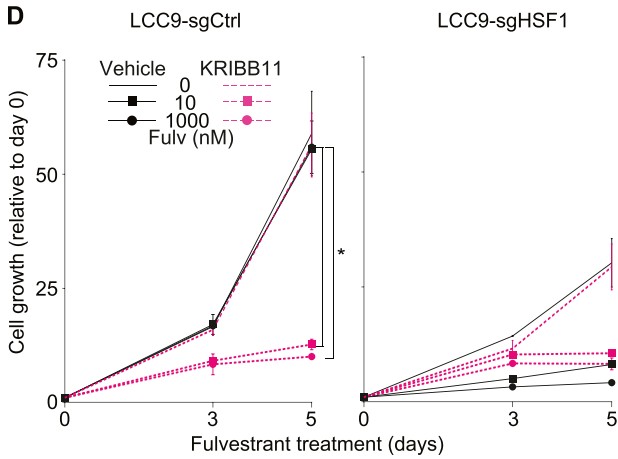

**Figure 6. The heat shock factor 1 (HSF1) inhibitor KRIBB11 restores response to fulvestrant in LCC9 cells.**
**(A)** Western blot analysis of HSF1 and estrogen receptor α protein levels in MCF7 cells overexpressing HSF1 (MCF7-HSF1) and control cells (MCF7-control) treated with KRIBB11. Cells were treated with 1 μM KRIBB11 for the indicated number of days before total protein extraction. Protein quantifications were performed

was the reason for the restoration of the antiestrogen response, we combined HSF1 depletion with fulvestrant treatments. As expected, fulvestrant led to degradation of the ERα in LCC9 depleted in HSF1 (Fig S3A). Accordingly, recruitment of the ERα, MED1, and BRD4 were decreased at ERα-activated genes after a 1-h fulvestrant treatment (Fig S3B and D). These results confirm that modulating HSF1 levels influence the ERα-dependent transcriptional program.

### Inhibition of HSF1 restores antiestrogen response in LCC9 cells

Because HSF1 is a pro-survival factor associated with resistance to treatment in multiple cancers (Dong et al, 2019, 2020), development of small-molecule inhibitors is providing the opportunity to test new combinatorial strategies to target resistant breast cancers. As a proof-of-concept, we wondered if inhibition of HSF1 would be sufficient to restore antiestrogen response in LCC9 cells. We selected the HSF1 inhibitor KRIBB11 which was shown to inhibit HSF1 activity (Yoon et al, 2011; Yallowitz et al, 2018). To validate the specificity of the KRIBB11 inhibitor in our experimental design, we tested its efficacy on the antiestrogen resistance observed after HSF1 overexpression in MCF7 cells (Figs 6A and B and S4A). Treatment of the cells with the inhibitor led to a small decrease in HSF1 levels in control and HSF1-overexpressing cells (Fig 6A). As expected, HSF1-overexpressing MCF7 cells maintained proliferation compared with control cells when treated with antiestrogens. However, when combinatorial fulvestrant-KRIBB11 (Fig 6B) or tamoxifen-KRIBB11 (Fig S4A) treatments were used, the proliferation of HSF1-overexpressing MCF7 cells was strongly reduced supporting that the KRIBB11 is a specific HSF1 inhibitor in our experimental design. Next, we wanted to determine if inhibition of HSF1 was sufficient to restore antiestrogen response in LCC9 cells. Interestingly, the KRIBB11 inhibitor did not affect HSF1 protein levels but slightly decreased the low ERα levels found in LCC9 cells (Fig 6C). These results are contrasting with the HSF1 depletion experiments (Figs 4 and 5), which led to an increase in ERα and suggest that reduced HSF1 levels and inhibition of its transcriptional activity differentially affect ERα. Nonetheless, whereas

using the ImageJ software and GAPDH was used as a loading control. **(B)** MCF7-HSF1 and MCF7-control cells were treated with the HSF1 inhibitor KRIBB11 (1 μM) and fulvestrant (Fulv). KRIBB11 was used alone or in combination with fulvestrant (10 and 1,000 nM) for the indicated number of days. MCF7-control cells are sensitive to fulvestrant, whereas MCF7-HSF1 cells are not. In presence of KRIBB11, proliferation of MCF7-HSF1 cells treated with fulvestrant is reduced. Cells were counted manually on days 0, 3, and 5 and values are represented as a fold compared with Day 0. Means ± SD (n = 2 biological replicates) are represented and *P*-values were calculated using a univariate *t* test. **(C)** Western blot analysis of HSF1 and estrogen receptor α protein levels in LCC9 cells depleted from HSF1 (LCC9-sgHSF1) and control cells (LCC9-sgCtrl) treated with KRIBB11. Cells were treated with 1 μM KRIBB11 for the indicated number of days before total protein extraction. Protein quantifications were performed using the ImageJ software and GAPDH was used as a loading control. **(D)** LCC9-sgHSF1 and LCC9-sgCtrl cells were treated with the HSF1 inhibitor KRIBB11 (1 μM) and fulvestrant (Fulv). KRIBB11 was used alone or in combination with fulvestrant (10 and 1,000 nM) for the indicated number of days. LCC9-sgCtrl cells are resistant to fulvestrant, whereas LCC9-sgHSF1 are not. In presence of KRIBB11, proliferation of LCC9-sgCtrl cells treated with fulvestrant is reduced. Cells were counted manually on days 0, 3, and 5 and values are represented as a fold compared with Day 0. Means ± SD (n = 2 biological replicates) are represented and *P*-values were calculated using a univariate *t* test. *\*P ≤ 0.05.*

KRIBB11 alone did not significantly affect the proliferation of LCC9 cells, proliferation of the cells treated with KRIBB11 and either fulvestrant (Fig 6D) or tamoxifen (Fig S4B) was strongly reduced to levels similar to HSF1-depleted LCC9 cells. Taken together, our results provide a proof-of-concept that inhibition of HSF1 in combination with antiestrogens is a promising combinatorial therapeutic approach for ERα-positive endocrine-resistant breast cancers.

# Discussion

In summary, we are proposing that overexpression of HSF1 in ERα-positive breast cancers is associated with a decrease dependency on the ERα-controlled transcriptional program for cancer growth. Accordingly, high HSF1 levels are associated with endocrine resistance and poor prognosis for breast cancer patients (Santagata et al, 2011; Mendillo et al, 2012; Gökmen-Polar & Badve, 2016). HSF1 is a pro-survival factor for which high expression levels are associated with resistance to treatment in multiple cancers (Dong et al, 2019). Further studies will be required to determine if overexpression of HSF1 is a general mechanism causing resistance to treatment.

# Materials and Methods

## Cell culture

MCF7 (HTB-22; ATCC), T47D (HTB-133; ATCC), and LCC1, LCC2, and LCC9 (provided by the laboratory of Robert Clarke) cells were grown in DMEM (#11965–092; Gibco) supplemented with 10% fetal bovine serum (#12483020; Invitrogen), 100 μM MEM nonessential amino acids (#25–0250; Cellgro), 2 mM L-glutamine (#25030–081; Gibco), 100 U/ml penicillin, and 100 μg/ml streptomycin (#15170–063; Gibco). Cells were treated with (Z)-4-Hydroxytamoxifen (#H7904; Sigma-Aldrich), Fulvestrant (ICI 182,780, #I4409; Sigma-Aldrich), KRIBB11 (#385570; Sigma-Aldrich), MG-132 (#C2211; Sigma-Aldrich), or vehicles as indicated. For heat shock, the cells were grown at 42°C for the indicated times.

## Western blot

Cells were harvested in cold PBS and homogenized in RIPA buffer (#RB4477; Biobasic) supplemented with protease inhibitors (#11697498001; Roche), for 10 min at 4°C. After centrifugation, proteins were quantified using a colorimetric assay (#5000111; Bio-Rad). Between 5 and 30 μg of total protein extracts were loaded. After transfer, membranes were blocked with a 5% solution of non-fat powdered milk or 5% bovine serum albumin proteins in TBS-0.1% Tween for a minimum of 1 h before antibody hybridization. The following primary antibodies were incubated overnight at 4°C: HSF1 (#4356, 1:1,000; Cell Signaling), ERα (#sc-543, 1:1,000; Santa Cruz), MED23 (#A300-425A, 1:1,000; Bethyl Laboratories), BRD4 (#A301-985A100, 1:1,000; Bethyl Laboratories), β-actin (MA5-15739, 1:5,000; Invitrogen), V5 Tag (#R960-25, 1:5,000; Invitrogen), Vinculin (V9131, 1:5,000; Sigma-Aldrich), and GAPDH (#MA5-15738, 1:5,000; Invitrogen).

Antirabbit (#111-035-003, 1:25,000; Jackson Immunoresearch Laboratories Inc.) and antimouse (#115-035-003, 1:25,000; Jackson Immunoresearch Laboratories Inc.) secondary antibodies coupled to the horseradish peroxidase were used for detection using luminescence (Clarity Western ECL Blotting Substrates, #1705060; Bio-Rad). Images were captured using a Chemidoc MP Image System (Bio-Rad) and quantified using the Image Lab Software.

## RNA levels

Cells were washed in cold PBS and harvested using the TriPure Isolation Reagent (# 11667157001; Sigma-Aldrich). RNA samples were purified using the GeneJET RNA Purification Kit (#K0732; Thermo Fisher Scientific) and quantified using a NanoDrop 2000 spectrophotometer. Samples were spiked (1 μl of a 1:5,000 dilution) with the ERCC RNA Spike-in-Mix (#4456740; Thermo Fisher Scientific) before reverse transcription using the SuperScript VILO Master Mix kit (#11755500; Thermo Fisher Scientific). For nascent transcripts, the Click-iT Nascent RNA Capture Kit (#C10365; Thermo Fisher Scientific) was used. Briefly, cells were treated with 0.5 mM 5-ethynyl-2′-deoxyuridine for the indicated times. Then, the cells were washed in cold PBS and RNA was extracted. After biotinylation, nascent RNA molecules were captured using streptavidin beads and used as a template for cDNA synthesis using the SuperScript VILO cDNA synthesis kit (#11754-050; Thermo Fisher Scientific). Real-time PCR were performed using the SYBR Select Master Mix (#4472920; Thermo Fisher Scientific). Primer sequences for each gene are available below:

**Primer sequences used for qRT-PCR**

| Gene name | Forward sequence | Reverse sequence |
|---|---|---|
| HSPD1 | AAGGAAGGCTTCGAGAAGATTAG | GGTCACAGGTTTAGACTGCTT |
| HSPH1 | GCACAGATTGTTGGCCTAAAC | CCACTATCCGAGGTTTCTCATC |
| HSP90AA1 | CTTGACCAATGACTGGGAAGAT | CACGTCGTGGGACAAATAGAA |
| HSF1 | AGCAGCTCCTTGAGAACATC | TGTCCTGGCGGATCTTTATG |
| TFF1 | CCCTCCCAGTGTGCAAATAAG | GGAGGGACGTCGATGGTATTA |
| GREB1 | CCCATCTTTTCCCAGCTGTA | ATTTGTTTCCAGCCCTCCTT |
| PGR | AGGTCTACCCGCCCTATCTC | CAAATCTTCTGAGGTAATGACTCG |
| ESR1 | CCAGGGAAGCTACTGTTTGC | TGATGTAGCCAGCAGCATGT |
| CCNG2 | GAACCTCCACAACAGCTACTAT | TCACAAGAGTCCTCACTTTCAC |
| MMD | GCAGCTGGAGGAACCATTTA | GGAGAGAATCCCATTGTGAGATAG |
| SMAD6 | CTGTCCGATTCCACATTGTCT | CATGCTGGCGTCTGAGAAT |
| ACTB | CGCGAGAAGATGACCCAGAT | AGAGGCGTACAGGGATAGCA |
| ERCC control | GGTCTGCTGACAAAGCATGA | TCGCCCCAGTAGTTTCTGTT |

## Chromatin immunoprecipitation (ChIP)

ChIP experiments were performed as described previously (Bilodeau et al, 2009; Kagey et al, 2010; Fournier et al, 2016; Boudaoud et al, 2017). Briefly, 50 million cells were cross-linked for 10 min with 1% formaldehyde and quenched with 125 mM glycine for 5 min. Cells were then washed twice with PBS, pelleted by centrifugation at 1,350g at 4°C for 5 min, flash-frozen, and stored at −80°C. After cellular lysis, pellets were sheared

between 200 and 600 bp using a Bioruptor Sonicator (Diagenode). Chromatin extracts from 15 to 20 million cells were immunoprecipitated overnight at 4°C with 50 $\mu$l of magnetic beads (#10004D; Life Technologies) saturated with 5 $\mu$g of antibodies. All ChIP-grade antibodies used were previously described: HSF1 (#4356; Cell Signaling) (Kourtis et al, 2015), ER$\alpha$ (#06-935; Sigma-Aldrich) (Glont et al, 2019), MED1 (#A300-793A; Bethyl Laboratories) (Kagey et al, 2010), and BRD4 (#A301-985A100; Bethyl Laboratories) (Lovén et al, 2013). After washes and reverse cross-link, DNA was purified using phenol extraction. ChIP-qPCR was performed using the SYBR Select Master Mix. The primer sequences for each genomic region are available below:

**Primer sequences used for ChIP-qPCR**

| Region | Forward sequence | Reverse sequence |
|---|---|---|
| HSPD1 (promoter) | TACGGCTCAAGGGTCAAATC | AAGGAGCTGTTTCTAGGCTTT |
| HSPH1 (promoter) | CAGCCTTATGTATCGCACTGA | AGAAGAAGGAAGCGGAAGTG |
| HSP90AA1 (promoter) | CTAAGTGACCGCACAGGA | CCCGTCGCTATATAAGGCA |
| TFF1 (promoter) | CCGAGTCAGGGATGAGAGG | GGCCTCCTTAGGCAAATGTT |
| GREB1 (enhancer) | GAGCTGACCTTGTGGTAGGC | CAGGGGCTGACAACTGAAAT |
| ESR1 (enhancer) | CTGCAGTAGGCACTCAGTAAAT | TCAAACTAACCTGAAACTCGGT |
| Control region #1 | ATGTCAGGCCCATGAACGAT | GCATTCATGGAGTCCAGGCTTT |
| Control region #2 | AGGACCTGCAGCAAACAGAA | TGTCTACATGGGCTAGTGTGCT |

### HSF1 gain-of-function

For gain-of-function experiments, the HSF1 (ccsbBroad304_00794) and ESR1 (ccsbBroad304_00517) vectors from the Horizon Discovery CCSB Human ORFeome Libraries and the empty vector (#25890; Addgene) were used. Briefly, vectors were transfected with the packaging plasmids psPAX2 (#12260; Addgene) and pMD2.G (#12259; Addgene) in HEK293T cells to produce lentiviral particles (Fournier et al, 2016). Supernatants were harvested 24 and 48 h after transfection, filtered, and added to recipient cell lines with 8 $\mu$g/ml of hexadimethrine bromide (#H9268; Sigma-Aldrich). Cells were transduced with lentiviral particles for 24 h before selection for 5–21 d. HSF1 protein levels were quantified by Western blot. To create the K302A, K303A ER$\alpha$ mutant, the wild-type ESR1 (ccsbBroad304_00517) was mutated using the Q5 site-directed mutagenesis kit (E0554S; New England Biolabs) with the following primers: 5′-CAAACGCTCTGCGGCGAACAGCCTGGCCTTGTC-3′ and 5′-ATCATGAGCGGGCTTGGC-3′. Lentiviruses were produced and transduced as described above.

### HSF1 loss-of-function

For the HSF1 loss-of-function experiments, CRISPR/Cas9 editing was used. Briefly, vectors carrying sgRNAs and GFP-Cas (LentiGuide-puro; #52963; Addgene) were transfected with the packaging plasmids psPAX2 (#12260; Addgene) and pMD2.G (#12259; Addgene) in HEK293T cells to produce lentiviral particles (Fournier et al, 2016). Supernatants were harvested 24 h after transfection, filtered, and added to recipient cell lines with 8 $\mu$g/ml of hexadimethrine bromide (#H9268; Sigma-Aldrich). Cells were transduced with lentiviral particles for 24 h before selection for 5–14 d. HSF1 protein levels were quantified by Western blot. Sequences for each gRNA are available below:

**Sequences of each gRNA used for loss-of-function experiments**

| Target | Forward sequence | Reverse sequence |
|---|---|---|
| HSF1 | caccgACTGGCCCTGG TCGAACACG | aaacCGTGTTCGACCAG GGCCAGTc |
| Control | caccgATCGTTTCCGC TTAACGGCG | aaacCGCCGTTAAGCG GAAACGATc |

### Cell proliferation and soft agar colony formation assays

For proliferation assays, cells were seeded into clear bottom 12-well plates and manually counted after the indicated number of days. Soft-agar colony formation assays were performed as previously described (Fournier et al, 2016). The protocol was adapted by using 1% Agar Noble as the bottom layer and 0.7% Agar Noble as the top layer. Briefly, clear-bottom six-well plates were seeded with 2,000 cells in DMEM supplemented with 10% fetal bovine serum, 100 $\mu$M MEM nonessential amino acids, 2 mM L-glutamine, 100 U/ml penicillin and 100 $\mu$g/ml streptomycin. Tamoxifen and fulvestrant were added in the agar layers and the culture media every 3 d at the indicated concentrations. Cells were maintained for 15 d before fixation, staining with Crystal Violet and quantification using an inverted microscope.

### Breast cancer data analysis

For gene expression changes in breast cancer, processed standardized expression datasets generated by the TCGA Research Network (https://www.cancer.gov/tcga) were used. A total of 459 samples were listed as positive for "breast_carcinoma_estrogen_receptor_status" (ER$\alpha$+) and were used in the analysis. The data were transformed from $\log_2$ to linear space. We then applied scaling at the gene level to account for the different basal expression for each gene (scale in R). Gene expression correlations between HSF1 and all other genes were calculated using the Pearson correlation function (cor under R). Genes with correlations above 0.3 and below –0.3 were kept for display. Data were clustered using the ward method for genes (columns) and we used HSF1 RNA levels for ranking (rows).

## Supplementary Information

## Acknowledgements

We would like to thank Marc Mendillo, Anne-Marie Pulichino, and members of the laboratory for helpful discussions and the critical review of the

manuscript. In addition, we would like to thank Michèle Fournier and Sofiane Yacine Mersaoui for technical advice and Robert Clarke for sharing the LCC cellular models. Last, we would like to thank Marc Mendillo and Mathieu Laplante for providing plasmids. This work was supported by funds from the Canada Research Chair in Transcriptional Genomics (grant #950-231582 to S Bilodeau), from the Natural Sciences and Engineering Research Council of Canada (grants #436266-2013 and #2019-06490 to S Bilodeau), the Canadian Institutes for Health Research (grant #MOP-126058 to S Bilodeau), the merit scholarship program for foreign students (to MAD Silveira), and doctoral research scholarships (to MAD Silveira and C Tav) from the Fonds de recherche du Québec.

## Author Contributions

MAD Silveira: conceptualization, formal analysis, investigation, methodology, writing—original draft, review, and editing.
C Tav: formal analysis.
F-A Bérubé-Simard: methodology.
T Cuppens: formal analysis.
M Leclercq: formal analysis.
É Fournier: formal analysis and methodology.
MC Côté: methodology.
A Droit: formal analysis.
S Bilodeau: conceptualization, supervision, funding acquisition, project administration, and writing—original draft, review, and editing.

## Conflict of Interest Statement

The authors declare that they have no conflict of interest.

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
