## [Reviewer comments · Life Science Alliance]

Life Science Alliance

Modulating HSF1 levels impacts expression of the estrogen receptor α and antiestrogen response

Maruhen Silveira, Christophe Tav, Félix-Antoine Bérubé-Simard, Tania Cuppens, Mikaël Leclercq, Éric Fournier, Maxime Côté, Arnaud Droit, and Steve Bilodeau

DOI: <https://doi.org/10.26508/lsa.202000811>

Corresponding author(s): Steve Bilodeau, Centre de Recherche du CHU de Québec - Université Laval

Review Timeline:

Submission Date:	2020-06-10
Editorial Decision:	2020-07-22
Appeal Requested:	2020-07-24
Editorial Decision:	2020-08-20
Revision Received:	2020-10-03
Editorial Decision:	2020-11-12
Revision Received:	2021-01-11
Editorial Decision:	2021-01-27
Revision Received:	2021-02-01
Accepted:	2021-02-03

Scientific Editor: Shachi Bhatt

Transaction Report:

July 22, 2020

Re: Life Science Alliance manuscript #LSA-2020-00811-T

Dr. Steve Bilodeau
Centre de Recherche du CHU de Québec - Université Laval
9 McMahon Street
Québec, Quebec G1R2J6
Canada

Dear Dr. Bilodeau,

Thank you for submitting your manuscript entitled "Modulating HSF1 levels impacts expression of the estrogen receptor α and antiestrogen response". The manuscript has been evaluated by expert reviewers, whose reports are appended below. Unfortunately, after an assessment of the reviewer feedback, we have decided that we cannot publish the dataset in Life Science Alliance in its current form.

While reviewers appreciate the study and find it in principle interesting, they also raise a number of substantive concerns which preclude publication of the paper, at least in its current form.

As you will see, both reviewers #1 and #3 feel that the conceptual advance of the current work is somewhat limited.

Providing further insight into the mechanisms by which the HSF1 and ER α pathways interact would bring the manuscript to level where its value to the community would be more clear. For example, ref 1 suggests to perform binding site analysis of HSF1 and MTA1 to show they are colocalized. Along the same lines, reviewer #2 recommend further experimental support for the claim that HSF-1 is a master regulator of the endocrine response (suggestion demonstration that over-expression of HSF-1 in wild type cells renders them insensitive to tamoxifen/fulvestrant).

Further, ref 1 in particular would expect the generality of the conclusions to be better demonstrated by including a genome-wide analysis of the cross-talk ('perform genome-wide binding analysis (ChIPseq) of ER α and HSF1 to assess the generality beyond three genes')

Reviewers #1 and #3 also suggest investigation of how HSF1 inhibition could be utilized in combination with the Tam/Fulv treatments.

In our view this is an alternative route to develop this dataset to make it suitable for publication in LSA. We would not make it a precondition to develop both angles.

The three reviewers raise a number of other related concerns which would need to be addressed before the paper could be considered suitable for publication in LSA. For example. ref 1 suggests to compare transcriptomes from ER α + breast cancer tissue and MCF7 and LCC9 endocrine-resistant cells.

Although your manuscript clearly represents a solid dataset and a good basis for an excellent paper, we feel that the points raised by the reviewers are more substantial than can typically be addressed in a reasonable revision period. If you wish to expedite publication of the current data, we would therefore understand if you preferred to pursue publication at another journal. However, if the suggested revisions are feasible, we would be happy to evaluate a suitably revised manuscript in

the future.

If you would like to resubmit this work to Life Science Alliance, please contact the journal office to discuss an appeal of this decision or you may submit an appeal directly through our manuscript submission system. Please note that priority and novelty would be reassessed at resubmission.

Regardless of how you choose to proceed, we hope that the comments below will prove constructive as your work progresses. We would be happy to discuss the reviewer comments further once you've had a chance to consider the points raised in this letter.

Thank you for thinking of Life Science Alliance as an appropriate place to publish your work.

Sincerely,

Reilly Lorenz
Editorial Office Life Science Alliance
Meyerhofstr. 1
69117 Heidelberg, Germany
t +49 6221 8891 414
e contact@life-science-alliance.org
www.life-science-alliance.org

Reviewer #1 (Comments to the Authors (Required)):

The manuscript by Silveira et al. investigates the modulatory effect of heat shock factor 1 (HSF1) on estrogen receptor α (ER α)-dependent transcriptional program that leads to antiestrogen resistance phenotype in breast cancer cells. The authors used well-characterized MCF7 cells-derived LCC cellular models (LCC1, LCC2, and LCC9) to explore the regulatory role of HSF1 in the antiestrogen response system. Overall, it is a straightforward study developing interesting observation, the key point being that overexpression of HSF1 negatively alters the transcriptional profile of ER α in endocrine-resistant breast cancer cells and its deletion restores anti-proliferative effects of antiestrogen. The authors envision that targeting HSF1 with small-molecule inhibitors in combination with hormonal therapies can have therapeutic benefits in endocrine-resistance cancer. However, they do not provide any mechanistic insight on how HSF1 and ER α pathways interact and what factors potentially link them. Further, since the major part of the study deals with transcriptional reprogramming related to respective transcription factors in endocrine resistance in the absence of genome-wide analysis of this cross-talk, the scope of the study is limited and in present format, provide no conceptual advancement over previously published observations.

Specific concerns:

1. The authors show overexpression of HSF1 in an endocrine-resistant LCC9 cell line and compare candidate gene activity related to HSF1 and ER α pathways in MCF7 and LCC9 cell lines. This has already been reported in multiple studies including cancer patient tissues, in mouse models in vivo as well as in cell lines (PMID: 22863008, 17922035, 29954368, 22042860, 27713164, etc.). To move this beyond repetition, the authors should compare transcriptomes from ER α + breast cancer tissue and MCF7 and LCC9 endocrine-resistant cells to gain insight into this cross-talk and how HSF1 affects ER α expression. Further, the authors should provide and experimentally establish the

rationale behind the competition between these two pathways that they propose.

2. The authors report ER α and HSF1 recruitment at their respective target genes in MCF7 and LCC9 cell lines using a small subset of genes (~2 fold change by ChIP-qPCR) and they show changes in occupancy of associated transcriptional co-regulators, MED1 and BRD4. If the hypothesis is that these two master regulators dictate the phenotype of the cells, the authors should perform genome-wide binding analysis (ChIPseq) of ER α and HSF1 to assess the generality beyond three genes. The authors should perform motif analysis to determine potentially overlapping binding sites, over-represented sites and novel sites appearing specifically in LCC9 cells.

3. The authors provide no experimental data to explain how HSF1 specifically blunts ER α pathway and what could be the associated factors driving this phenotype. The previous study (PMID: 17922035) shows that HSF1 represses ER α pathways in cooperation with MTA1, are there any other factors related to HSF1 function in this context? Also, the authors should perform binding site analysis of HSF1 and MTA1 to show they are colocalized. Again, in the absence of mechanistic insight - there is little conceptual development.

4. To corroborate these limited in vitro studies, the author should assess the combinatorial effect of HSF1 inhibitor and Tam/Fulv in endocrine-resistant cancer cells or in the mouse model (HSF1 $^{-/-}$), treated with Tam/Fulv to show restoration of anti-estrogen response.

Additional Points:

1. Fig 1A. The description of the quantification of western blots is unclear: how are the ratios derived and related to B-actin? Also, according to quantification the HSF1: ER α of LCC9 should be 9.3??

2. Fig 1B and 3C. The details of binding region of HSF1 target genes in ChIP-qPCR are not provided. In addition, the authors perform the univariate test for statistical significance, however, it is unclear how univariate t-test will be applied when two variables are being compared. The authors should perform a normal bivariate t-test.

3. Fig 1E. The authors should provide levels of expression of these genes rather than 'fold change' which disregards potentially different baselines. Same applies to Fig. S2.

4. Fig 2B and 3A. The authors show the 'ratio of ratios' of mRNA expression and perform an univariate t-test. This kind of data are NOT normally distributed which violates test assumptions. The authors should show house-keeping normalized expression levels for both conditions and perform a bivariate t-test.

5. Fig 3. HSF1 deletion may affect many pathways affecting ER indirectly. Complementation/rescue experiments with WT HSF1 need to be performed.

Reviewer #2 (Comments to the Authors (Required)):

This manuscript explores the role of HSF-1 in endocrine resistance, with the authors showing that

in isogenically matched wild type and resistant MCF7 derivatives, HSF-1 levels increase, whereas ER levels and binding decreases. The authors show that depletion of HSF-1 in resistant lines can re-instate response to ER antagonists. Experiments are conducted to map ER components or HSF-1 on gene targets of ER or HSF-1, which is correlated with MED1 binding. Data is provided to show that HSF-1 is critical for the resistance and when depleted, results in active ER transcription, as measured by a handful of ER target genes and ER binding to regulatory elements.

This is a solid paper, the results on HSF-1 are convincing and clear and the paper and figures are presented nicely.

- Suggesting that HSF-1 is the master transcription factor that dictates endocrine response, is a substantial claim. To show this the authors need to show that over-expression of HSF-1 in wild type cells renders them insensitive to tamoxifen/fulvestrant. Removing it from resistant lines shows that it plays a role, but this doesn't implicate it as the cause. Whilst this is a big experiment and I am not necessarily expecting the authors to conduct this, this would be expected if they want to keep the text as it currently is, stating that HSF-1 is the purported master and catalytic transcription factor. I suspect it might be better if the authors modify their text to tone down the magnitude of the claim.

- The use of one gRNA is problematic, given the variability between gRNAs. Have the authors managed to validate one or two of the key figures with other gRNAs or other reagents (siRNA/shRNA etc?).

Reviewer #3 (Comments to the Authors (Required)):

In the present manuscript Silveira et al. presented evidence that low levels of stress response master regulator HSF1 is associated with breast cancer resistance. Reducing HSF1 re-establishes ER α levels along with the response to antiandrogens. The manuscript is interesting and studies a relevant issue in the treatment of resistant breast cancers, for which the treatments available are not sufficiently efficient to guarantee the survival of these patients. The experiments are well carried out and addressed the proposed questions.

However, the strategy of ER α re-expression for the treatment of breast cancer is not novel and there is a large literature on it. DNA methylation and chromatin remodeling are two epigenetic mechanisms that have been linked with the lack of ER α expression in breast cancers. The demethylation of the ER α promoter or treatment with HDAC inhibitors shows promise in restoring ER α expression in ER- breast cancers. Moreover, inhibition of MAPK activity can cause re-expression of the ER α and restore sensitivity to endocrine therapies (Brinkman JA et al., 2009 J Mammary Gland Biol Neoplasia)

The question that arises is whether HSF1 is associated with any of the mechanisms mentioned above. Moreover, from a mechanistic point of view: how HSF1 regulates ER α expression? Is HSF1 bound at/close ER α gene in LCC9 cells? In case it is, is HSF1 associated with corepressors that maintain defined histone acetylation or methylation patterns at the ER α gene reducing concomitantly its expression? In the same lines, how HSF1 depletion changed ER α expression? or HSF1 knockdown involved a more indirect effect? Authors should clarify these points.

A second point that needs further development is how HSF1 targeting could be a realistic complementary strategy of current treatments such as Fulvestrant or Tamoxifen.

Dear Reilly,

Thank you for providing comments and suggestions that will greatly enhance the quality of our manuscript. After carefully considering the points raised by the reviewers, we believe that we can address most of them. Indeed, we were following up on most of these avenues to broaden the scope of the manuscript. Attached, we are providing a point-by-point response to the reviewers' comments (in blue are available data/information; in red are experiments that could be done in a reasonable 4-6 weeks time frame).

In the next version of the manuscript, we will increase the scope of the study and provide new and important conceptual advancements. Mainly,

- we will establish that HSF1 overexpression is sufficient to trigger endocrine resistance.
- we will provide evidence that HSF1 interferes with ERα expression levels post-transcriptionally.
- we will show that HSF1 inhibitors in combination with tamoxifen and fulvestrant are a potential avenue for treating resistant breast cancer tumors.
- we are hoping to demonstrate that HSF1 overexpression levels in breast cancer patients are anticorrelated with a global decrease in the ERα-controlled gene expression program.

Therefore, we feel that we will be broadly answering most comments and suggestions. Hopefully, our proposition would make the manuscript appropriate for publication in Life Science Alliance.

Please let us know if our plan is reasonable and if we should consider moving forward with a revised version of the manuscript.

Best Regards,

Steve

Steve Bilodeau, Ph.D.
Canada Research Chair in Transcriptional Genomics
Associate Professor, Faculty of Medicine, Université Laval
Principal Investigator, Centre de Recherche du CHU de Québec à l'Université Laval
Principal Investigator, Centre de recherche sur le Cancer de l'Université Laval
Principal Investigator, Centre de recherche en données massives de l'Université Laval
9 McMahon Street
Québec City, Canada
G1R 2J6
Phone: (418) 525-4444 ext.15550
Fax: (418) 691-5439

Lab Website: Bilodeau laboratory - CRC
Publications: Research Gate
Networking: LinkedIn

August 20, 2020

MS: LSA-2020-00811-T

Dr. Steve Bilodeau
Centre de Recherche du CHU de Québec - Université Laval
9 McMahon Street
Québec, Quebec G1R2J6
Canada

Dear Dr. Bilodeau,

Thank you for submitting an appeal for your manuscript "Modulating HSF1 levels impacts expression of the estrogen receptor α and antiestrogen response" [LSA-2020-00811].

We have evaluated your point-by-point response, appeal letter and re-evaluated the concerns raised by the reviewers. Based upon your response, we agree to send the revised manuscript (revised according to the plan laid out in the pbp letter) back to the referees.

We are slightly concerned that some of the referees' requests won't be fully addressed (for eg. genome wide CHIP-seq, validation using second gRNA or siRNA/shRNA and the correlative nature of the findings showing how HSF1 affects ER α expression), but nevertheless, since the plan laid out in the appeal letter does address a number of referees' concerns, we would be happy to send it back to the referees for re-review. We usually try to keep the same reviewers for re-review.

Please use the following link to submit your manuscript when ready:

<https://lsa.msubmit.net/cgi-bin/main.plex?el=A6Na6VL3A3CjjL4I5B9ftdp8QIAGiDWWsaGJuyONerwZ>

Yours sincerely,

Shachi Bhatt
Executive Editor
Life Science Alliance

Response to Reviewer #1 :

The manuscript by Silveira et al. investigates the modulatory effect of heat shock factor 1 (HSF1) on estrogen receptor α (ER α)-dependent transcriptional program that leads to antiestrogen resistance phenotype in breast cancer cells. The authors used well-characterized MCF7 cells-derived LCC cellular models (LCC1, LCC2, and LCC9) to explore the regulatory role of HSF1 in the antiestrogen response system. Overall, it is a straightforward study developing interesting observation, the key point being that overexpression of HSF1 negatively alters the transcriptional profile of ER α in endocrine-resistant breast cancer cells and its deletion restores anti-proliferative effects of antiestrogen. The authors envision that targeting HSF1 with small-molecule inhibitors in combination with hormonal therapies can have therapeutic benefits in endocrine-resistance cancer. However, they do not provide any mechanistic insight on how HSF1 and ER α pathways interact and what factors potentially link them. Further, since the major part of the study deals with transcriptional reprogramming related to respective transcription factors in endocrine resistance in the absence of genome-wide analysis of this cross-talk, the scope of the study is limited and in present format, provide no conceptual advancement over previously published observations.

We thank the reviewer for the comments and suggestions. We are now increasing the scope of the analysis to provide new and important conceptual advancements. The manuscript was expanded from 3 to 6 main figures to address multiple points. First, we are establishing that HSF1 overexpression is sufficient to trigger partial antiestrogen resistance in MCF7 cells (Fig. 1). Second, we are providing evidence that HSF1 destabilizes ER α protein levels prior to transcriptional consequences (Fig. 2). Third, analysis of TCGA data established that HSF1 overexpression levels in breast cancer patients are associated with a global decrease in the ER α -controlled gene expression program (Supplementary Fig. S2). Fourth, we are establishing that inhibition of HSF1 in combination with tamoxifen and fulvestrant is a potential avenue for treating endocrine resistant breast cancers (Fig. 6 and Supplementary Fig. S4). Therefore, we believe that this improved version of the manuscript is a significant conceptual advancement over previously published observations.

Specific concerns:

1. The authors show overexpression of HSF1 in an endocrine-resistant LCC9 cell line and compare candidate gene activity related to HSF1 and ER α pathways in MCF7 and LCC9 cell lines. This has already been reported in multiple studies including cancer patient tissues, in mouse models in vivo as well as in cell lines (PMID: 22863008, 17922035, 29954368, 22042860, 27713164, etc.).

The reviewer is right that HSF1 overexpression is associated with poor prognosis for breast cancer patients and repression of the ER α -dependent gene expression program. We emphasize those findings in the introduction as the rationale to conduct the study. As the reviewer pointed out, in the original submission the innovation relied on

“overexpression of HSF1 negatively alters the transcriptional profile of ER α in endocrine-resistant breast cancer cells and its deletion restores anti-proliferative effects of antiestrogen”. As described above, in this new version, we go beyond this original discovery.

To move this beyond repetition, the authors should compare transcriptomes from ER α + breast cancer tissue and MCF7 and LCC9 endocrine-resistant cells to gain insight into this cross-talk and how HSF1 affects ER α expression.

To gain insights into the global cross-talk between HSF1 and ER α , we compared the gene signatures controlled by each transcription factor to the transcriptome of ER α + breast cancers. We collected 459 datasets (listed as ER α +) from TCGA. Then, we measured pairwise correlations between HSF1 and each gene. Next, we cross-referenced with previously validated gene signatures for HSF1 (Mendillo et al. 2012) and ER α (Abba et al. 2005). As expected, the HSF1-regulated genes were found to correlate with HSF1 levels. We identified 97 genes with correlations higher or lower than a 0.3 threshold. Indeed, 47 HSF1-regulated genes were positively correlated with HSF1 for only 7 that were anticorrelated. For ER α -regulated genes, 34 were anticorrelated for only 8 that were positively correlated. Therefore, we feel confident that our conclusions is valid: high levels of HSF1 are associated with a decrease of the ER α -dependent gene expression program in breast cancer. We created Supplementary Fig. S2, updated the main text and the materials and methods. While some LCC9 RNA-seq datasets were available (GSE55343, GSE54891), their low quality prohibited their comparison with the TCGA datasets.

Further, the authors should provide and experimentally establish the rationale behind the competition between these two pathways that they propose.

This is a valid and important point. First, we aimed to establish that HSF1 was sufficient to induce an antiestrogen resistance phenotype. HSF1 overexpression decreased ER α protein levels (Fig. 1A) corroborating the increase of ER α in HSF1-depleted LCC9 cells (Fig. 5B). Similarly, RNA levels of ER α and known target genes were downregulated 5 days after HSF1 transduction (Fig. 1B). These gene expression changes were associated with partial resistance to tamoxifen and fulvestrant (Fig. 1C-F).

Next, we wanted to decipher if the primary effect of HSF1 was transcriptional or not. We investigated recruitment of HSF1 by ChIP and could not detect recruitment at ER α -regulated genes in MCF7 or LCC9 cells. To validate that HSF1 was not recruited to ER α -regulated genes, we turned to heat shock as it triggers a massive activation of HSF1 (Fig. 2A-C). No recruitment of HSF1 was detected at ER α -regulated genes during the course of the experiment. Interestingly, the heat shock experiment was informative on the timing. Indeed, the decrease in ER α protein levels was observed after 15-20 minutes (Fig. 2A). However, the effect on transcription was seen only at 60 minutes (Fig. 2B). Furthermore, overexpression of HSF1 in MCF7 cells did not lead to increased recruitment

at ER-regulated genes (Fig. 2D). These results suggest that ER α is degraded prior to observe transcriptional changes at regulated genes.

To support the model that ER α protein levels are destabilized by HSF1, MCF7 cells overexpressing HSF1 and their matching control were treated with the proteasome inhibitor MG-132 (Fig. 2E). Following a 4 hours MG-132 treatment, ER α levels were increased by 40% supporting its constant turnover in MCF7 cells. As expected, overexpression of HSF1 led to 50% decrease in ER α protein levels. However, when HSF1-overexpressing cells were treated with MG-132, ER α levels recovered to level similar to control cells. Therefore, the results support the role of HSF1 in destabilizing ER α protein levels.

The text was updated to include these new figures.

2. The authors report ER α and HSF1 recruitment at their respective target genes in MCF7 and LCC9 cell lines using a small subset of genes (~2 fold change by ChIP-qPCR) and they show changes in occupancy of associated transcriptional co-regulators, MED1 and BRD4. If the hypothesis is that these two master regulators dictate the phenotype of the cells, the authors should perform genome-wide binding analysis (ChIPseq) of ER α and HSF1 to assess the generality beyond three genes. The authors should perform motif analysis to determine potentially overlapping binding sites, over-represented sites and novel sites appearing specifically in LCC9 cells.

Our model is that ER α and HSF1 are controlling two independent transcriptional programs that are both contributing to the proliferation of breast cancer cells. When cells express high levels of HSF1, ER α levels are decreased and the cells become less dependent on estrogens for growth. Therefore, we are not expecting major changes in the ER α and HSF1 cistromes, but rather a modulation of transcription factor recruitment at regulatory elements matching their variation in expression. This is why we decided to use ChIP-qPCR rather than ChIP-seq which are much more difficult to properly normalize for quantitative studies. Nonetheless, we ran a preliminary analysis for ER α binding in MCF7 compared to LCC9 cells. First, we compared ER α occupancy in MCF7 and LCC9 cells. The results showed the same ER α binding pattern, but with lower signal in LCC9 compared to MCF7 cells. Second, we compared ER α - and HSF1-occupied regions in MCF7 cells. The results showed a small 5% overlap even when HSF1 was activated by heat shock supporting that both transcription factors control independent transcriptional programs. Based on these preliminary data, we decided that generating a complete dataset would not improve the current model and focused on suggestions that would push it further.

3. The authors provide no experimental data to explain how HSF1 specifically blunts ER α pathway and what could be the associated factors driving this phenotype. The previous study (PMID: 17922035) shows that HSF1 represses ER α pathways in cooperation with MTA1, are there any other factors related to HSF1 function in this context? Also, the authors should

perform binding site analysis of HSF1 and MTA1 to show they are colocalized. Again, in the absence of mechanistic insight - there is little conceptual development.

Once again the reviewer raised an excellent point and model that we wanted to investigate. As described above, our model now proposes that activation of HSF1 first leads to a decrease in ER α protein. Furthermore, we could not detect HSF1 binding at ER α -regulated genes in LCC9 cells (data not shown), in MCF7 overexpressing HSF1 (Fig. 2D) and heat-shock stimulated MCF7 cells (Fig. 2C). Therefore, we do not believe the primary mechanisms to explain the phenotype involves HSF1-dependent recruitment of the NuRD complex at ER α -regulated genes. The text was updated to account with our new model.

4. To corroborate these limited in vitro studies, the author should assess the combinatorial effect of HSF1 inhibitor and Tam/Fulv in endocrine-resistant cancer cells or in the mouse model (HSF1 $^{-/-}$), treated with Tam/Fulv to show restoration of anti-estrogen response.

We agree with the reviewer that proving that HSF1 inhibitors could be used on endocrine-resistant cancer cells to restore Tam/Fulv response would be a very interesting corroboration. We selected the HSF1 inhibitor KRIBB11 since it was the best characterized and readily available. We validated the specificity of the KRIBB11 inhibitor with the HSF1 overexpression in MCF7 cells (Fig. 6 and Supplementary Fig. S4). The gain in proliferation observed in HSF1-overexpressing cells was lost when a combinatorial antiestrogen – KRIBB11 treatment was applied (Fig. 6A and Supplementary Fig. S4A). Therefore, these results support that the KRIBB11 is a specific HSF1 inhibitor in our experimental design. Next, we wanted to determine if inhibition of HSF1 would be sufficient to restore antiestrogen response in LCC9 cells. Again, KRIBB11 alone did not affect the proliferation of LCC9 cells. However, when LCC9 cells were treated with KRIBB11 and antiestrogens, proliferation slowed down to levels similar to HSF1-depleted LCC9 cells (Fig. 6B and Supplementary Fig. S4B). Taken together, our results provide a proof-of-concept that inhibition of HSF1 in combination with antiestrogens is a promising combinatorial therapeutic approach for ER α -positive endocrine-resistant breast cancers.

Additional Points:

1. Fig 1A. The description of the quantification of western blots is unclear: how are the ratios derived and related to B-actin? Also, according to quantification the HSF1: ER α of LCC9 should be 9.3??

We have updated the methods and figure legends to clearly explain the quantification methods used for each western blot throughout the manuscript. We thank the reviewer for picking up the error. It is now corrected.

2. Fig 1B and 3C. The details of binding region of HSF1 target genes in CHIP-qPCR are not provided. In addition, the authors perform the univariate test for statistical significance, however,

it is unclear how univariate t-test will be applied when two variables are being compared. The authors should perform a normal bivariate t-test.

The details of binding regions for HSF1 were added to the figure legend and methods. The statistical tests were updated to bivariate t-tests.

3. Fig 1E. The authors should provide levels of expression of these genes rather than 'fold change' which disregards potentially different baselines. Same applies to Fig. S2.

We disagree with the reviewer that an absolute quantification is more appropriate. Yes, the genes represented in the old Fig. 1E and Fig. S2 have different baselines, but the key results that we want to convey is the relative expression between MCF7 and LCC9 cells. We tried different versions and this is our best representation.

4. Fig 2B and 3A. The authors show the 'ratio of ratios' of mRNA expression and perform an univariate t-test. This kind of data are NOT normally distributed which violates test assumptions. The authors should show house-keeping normalized expression levels for both conditions and perform a bivariate t-test.

We agree with the reviewer and updated to bivariate t-tests to calculate significance. For the visual representation, we feel that relative expression is the best way to convey the message of gene expression changes for multiple genes. Therefore, we are presenting house-keeping normalized expression levels relative to the control value on a log₂ scale in Fig. 1B, 4B and 5A. Presenting the data on a linear scale with both conditions did not improve the visualization. The figure legends were updated.

5. Fig 3. HSF1 deletion may affect many pathways affecting ER indirectly. Complementation/rescue experiments with WT HSF1 need to be performed.

We are confused by the reviewer's suggestion. A complementation experiment would confirm the specificity of the HSF1 depletion experiment, but would not provide any information about pathways affecting ER indirectly. However, in this revised version, we are confirming that HSF1 overexpression is sufficient to induce partial antiestrogen resistance (Fig. 1) and that the HSF1 inhibitor KRIB11 is able to restore antiestrogen response in LCC9 cells (Fig. 6). Therefore, using these alternative approaches, we are confirming the direct implication of HSF1.

Response to Reviewer #2

This manuscript explores the role of HSF-1 in endocrine resistance, with the authors showing that in isogenically matched wild type and resistant MCF7 derivatives, HSF-1 levels increase, whereas ER levels and binding decreases. The authors show that depletion of HSF-1 in resistant lines can re-instate response to ER antagonists. Experiments are conducted to map ER components or HSF-1 on gene targets of ER or HSF-1, which is correlated with MED1 binding. Data is provided to show that HSF-1 is critical for the resistance and when depleted,

results in active ER transcription, as measured by a handful of ER target genes and ER binding to regulatory elements.

This is a solid paper, the results on HSF-1 are convincing and clear and the paper and figures are presented nicely.

We are pleased that the reviewer shares our enthusiasm for the manuscript. Answers to the specific comments are provided below.

- Suggesting that HSF-1 is the master transcription factor that dictates endocrine response, is a substantial claim. To show this the authors need to show that over-expression of HSF-1 in wild type cells renders them insensitive to tamoxifen/fulvestrant. Removing it from resistant lines shows that it plays a role, but this doesn't implicate it as the cause. Whilst this is a big experiment and I am not necessarily expecting the authors to conduct this, this would be expected if they want to keep the text as it currently is, stating that HSF-1 is the purported master and catalytic transcription factor. I suspect it might be better if the authors modify their text to tone down the magnitude of the claim.

We thank the reviewer for the comments. We now provide evidence that overexpression of HSF1 alone is sufficient to induce endocrine resistance in MCF7 and T47D cells (Fig. 1 and Supplementary Fig. S1). Furthermore, inhibition of HSF1 using KRIBB11 was sufficient to restore tamoxifen and fulvestrant responses in LCC9 cells (Fig. 6). Therefore, we feel confident that HSF1 is sufficient to lead toward an endocrine resistance phenotype.

However, claiming that HSF1 is the master transcription factor that dictates the endocrine response would be undercutting the work of several labs showing that multiple transcription factors/pathways are involved. We clarified the text to make explicit that HSF1 is an additional factor and not the only mechanism leading toward endocrine resistance.

- The use of one gRNA is problematic, given the variability between gRNAs. Have the authors managed to validate one or two of the key figures with other gRNAs or other reagents (siRNA/shRNA etc?).

In this revised version, we are providing two additional lines of evidence using different reagents. First, our results showed that overexpression of HSF1 is sufficient to trigger partial antiestrogen resistance (Fig. 1). Second, we are demonstrating that inhibiting HSF1 with the small molecule inhibitor KRIB11 restored antiestrogen response in LCC9 cells (Fig. 6). We agree with the reviewer that gRNA and shRNA sometimes lead to off target effects, but we feel that the probability is offset by our new data.

Response to Reviewer #3 :

In the present manuscript Silveira et al. presented evidence that low levels of stress response master regulator HSF1 is associated with breast cancer resistance. Reducing HSF1 re-

establishes ER α levels along with the response to antiandrogens. The manuscript is interesting and studies a relevant issue in the treatment of resistant breast cancers, for which the treatments available are not sufficiently efficient to guarantee the survival of these patients. The experiments are well carried out and addressed the proposed questions.

We are thankful for the careful revision of the manuscript and excited that the reviewer found our work highly relevant for the treatment of resistant breast cancer. We are addressing the specific points below.

However, the strategy of ER α re-expression for the treatment of breast cancer is not novel and there is a large literature on it. DNA methylation and chromatin remodeling are two epigenetic mechanisms that have been linked with the lack of ER α expression in breast cancers. The demethylation of the ER α promoter or treatment with HDAC inhibitors shows promise in restoring ER α expression in ER- breast cancers. Moreover, inhibition of MAPK activity can cause re-expression of the ER α and restore sensitivity to endocrine therapies (Brinkman JA et al., 2009 J Mammary Gland Biol Neoplasia).

We agree with the reviewers that multiple epigenetic and transcriptional mechanisms have been involved in the silencing of ER α expression and targeted to restore expression and sensitivity to antiestrogens. We are now increasing the scope of the analysis to provide new and important conceptual advancements. The manuscript was expanded from 3 to 6 main figures to address multiple points. First, we are establishing that HSF1 overexpression is sufficient to trigger partial antiestrogen resistance in MCF7 cells (Fig. 1). Second, we are providing evidence that HSF1 destabilizes ER α protein levels prior to transcriptional consequences (Fig. 2). Third, analysis of TCGA data established that HSF1 overexpression levels in breast cancer patients are associated with a global decrease in the ER α -controlled gene expression program (Supplementary Fig. S2). Fourth, we are establishing that inhibition of HSF1 in combination with tamoxifen and fulvestrant is a potential avenue for treating endocrine resistant breast cancers (Fig. 6 and Supplementary Fig. S4). Therefore, we believe that this improved version of the manuscript is a significant conceptual advancement over previously published observations.

The question that arises is whether HSF1 is associated with any of the mechanisms mentioned above. Moreover, from a mechanistic point of view: how HSF1 regulates ER α expression?

This is a valid and important point. First, we aimed to establish that HSF1 was sufficient to induce an antiestrogen resistance phenotype. HSF1 overexpression decreased ER α protein levels (Fig. 1A) corroborating the increase of ER α in HSF1-depleted LCC9 cells (Fig. 5B). Similarly, RNA levels of ER α and known target genes were downregulated 5 days after HSF1 transduction (Fig. 1B). These gene expression changes were associated with partial resistance to tamoxifen and fulvestrant (Fig. 1C-F).

Next, we wanted to decipher if the primary effect of HSF1 was transcriptional or not. We investigated recruitment of HSF1 by ChIP and could not detect recruitment at ER α -

regulated genes in MCF7 or LCC9 cells. To validate that HSF1 was not recruited to ER α -regulated genes, we turned to heat shock as it triggers a massive activation of HSF1 (Fig. 2A-C). No recruitment of HSF1 was detected at ER α -regulated genes during the course of the experiment. Interestingly, the heat shock experiment was informative on the timing. Indeed, the decrease in ER α protein levels was observed after 15-20 minutes (Fig. 2A). However, the effect on transcription was seen only at 60 minutes (Fig. 2B). Furthermore, overexpression of HSF1 in MCF7 cells did not lead to increased recruitment at ER-regulated genes (Fig. 2D). These results suggest that ER α is degraded prior to observe transcriptional changes at regulated genes.

To support the model that ER α protein levels are destabilized by HSF1, MCF7 cells overexpressing HSF1 and their matching control were treated with the proteasome inhibitor MG-132 (Fig. 2E). Following a 4 hours MG-132 treatment, ER α levels were increased by 40% supporting its constant turnover in MCF7 cells. As expected, overexpression of HSF1 led to 50% decrease in ER α protein levels. However, when HSF1-overexpressing cells were treated with MG-132, ER α levels recovered to level similar to control cells. Therefore, the results support the role of HSF1 in destabilizing ER α protein levels.

The text was updated to include these new figures.

Is HSF1 bound at/close ER α gene in LCC9 cells? In case it is, is HSF1 associated with corepressors that maintain defined histone acetylation or methylation patterns at the ER α gene reducing concomitantly its expression?

We were unable to detect a significant amount of HSF1 at the regulatory regions of ER α -controlled genes in LCC9 cells. Furthermore, we did not detect HSF1 at these regulatory regions when MCF7 were treated with heat shock or overexpressing HSF1 (Fig. 2C-D). Therefore, we did not pursue a direct role of HSF1 in the repression of ER α -regulated genes.

In the same lines, how HSF1 depletion changed ER α expression? or HSF1 knockdown involved a more indirect effect? Authors should clarify these points.

The reviewer is raising an interesting point. We pursued multiple lines of enquiry leading to the model described above. The text was updated to include these results and clarify the proposed model.

A second point that needs further development is how HSF1 targeting could be a realistic complementary strategy of current treatments such as Fulvestrant or Tamoxifen.

We agree with the reviewer that proving that targeting HSF1 could be a realistic complementary strategy to current treatments of endocrine-resistant cancers would be very interesting. We selected the HSF1 inhibitor KRIBB11 since it was the best characterized and readily available. We validated the specificity of the KRIBB11 inhibitor with the HSF1 overexpression in MCF7 cells (Fig. 6 and Supplementary Fig. S4). The gain

in proliferation observed in HSF1-overexpressing cells was lost when a combinatorial antiestrogen – KRIBB11 treatment was applied (Fig. 6A and Supplementary Fig. S4A). Therefore, these results support that the KRIBB11 is a specific HSF1 inhibitor in our experimental design. Next, we wanted to determine if inhibition of HSF1 would be sufficient to restore antiestrogen response in LCC9 cells. Again, KRIBB11 alone did not affect the proliferation of LCC9 cells. However, when LCC9 cells were treated with KRIBB11 and antiestrogens, proliferation slowed down to levels similar to HSF1-depleted LCC9 cells (Fig. 6B and Supplementary Fig. S4B). Taken together, our results provide a proof-of-concept that inhibition of HSF1 in combination with antiestrogens is a promising combinatorial therapeutic approach for ER α -positive endocrine-resistant breast cancers.

November 12, 2020

Re: Life Science Alliance manuscript #LSA-2020-00811-TR-A

Dr. Steve Bilodeau
Centre de Recherche du CHU de Québec - Université Laval
9 McMahon Street
Québec, Quebec G1R2J6
Canada

Dear Dr. Bilodeau,

Thank you for submitting your revised manuscript entitled "Modulating HSF1 levels impacts expression of the estrogen receptor α and antiestrogen response" to Life Science Alliance. The manuscript has been seen by the original reviewers whose comments are appended below. While the reviewers continue to be overall positive about the work in terms of its suitability for Life Science Alliance (LSA), some important issues remain.

As you will note from the reviews below, Reviewer 1 and 3 still have some concerns about the HSF-1 overexpression and proteosomal degradation of ERalpha - these would need to be addressed for further consideration at LSA. Reviewer 1 has listed out these concerns in their review, while Reviewer 3 also suggested the same and agreed with Reviewer 1 in personal communication with us, post submission of their review comments. Additionally, Rev 3 also raises a number of minor concerns that also need to be addressed in a revised manuscript.

Our general policy is that papers are considered through only one revision cycle; however, given that both Reviewer 1 and 3 agree that the data needs to be strengthened to support the claims in the manuscript, we are open to one additional short round of revision, should you chose to opt for it. Depending on the new data, we will first try to make a final decision on the re-submission in-house with the help of one of our academic editors or our advisory editors; if for some reason that is not possible, we might reach out to the original referees one more time.

Please let me know if you are open to one final round of revision for this manuscript. At resubmission, please also submit a letter that includes a point by point response to the remaining reviewer comments.

B. MANUSCRIPT ORGANIZATION AND FORMATTING:

Sincerely,

Shachi Bhatt, Ph.D.
Executive Editor
Life Science Alliance
<https://www.lsjournal.org/>
Tweet @SciBhatt @LSAJournal

Reviewer #1 (Comments to the Authors (Required)):

The manuscript by Silveira et al. investigates the modulatory effect of heat shock factor 1 (HSF1) on estrogen receptor α (ER α)-dependent transcriptional program that leads to anti-estrogen resistance phenotype in breast cancer cells. In the revised version of the manuscript, the authors propose that overexpression of HSF1 triggers proteasomal degradation of ER α , negatively impacting ER α -dependent gene regulation. Further, the authors used HSF1 inhibitor KRIBB11 in addition to Fulv/TAM to show that anti-estrogen response is restored in ER α + endocrine-resistant breast cancers. However, some of the new experiments require clarification and additional experiments.

1. One of the main conclusions of the study mechanistically is that overexpression of HSF1 leads to proteasomal degradation of ER α protein, thereby ablating its gene regulatory program. As evidence, the authors show that in response to HSF1 induction, ER α protein is degraded prior to cessation of proposed ER-mediated transcription. Further, in Fig 2E, ER α protein level is restored upon use of proteasome inhibitor MG-132. To connect the dots in a more than correlative fashion, the authors should show that treatment with proteasome inhibitor MG-132 restores ER α -dependent transcription program. Along the same lines, to conclusively establish that HSF1 triggers proteasomal degradation of ER α , the authors should repeat the experiments with ER mutated at Lysine 302 and 303 (PMID: 18388150) which would be expected to be resistant.

2. Further, it would be interesting to see how the dynamics of ER α binding at the respective genes change with HSF1 induction due to heat shock at different timepoints. The authors propose a regulatory mechanism of HSF1 negating ER α -dependent gene expression contrasting to the one proposed by Khalque et al 2008 (direct binding of HSF1 to ER α -target genes), these need to be addressed in detail. Also, does the induction of HSF1 in MCF7 cells upon heat shock lead to a similar transcriptional program as endocrine-resistant LCC9 cells or is there a difference in the regulatory mechanism?

3. Fig S2. The authors should describe how subset of HSF1 (97) and ER α (42) signature genes were selected from a much larger pool of HSF1/ER α regulated genes from the dataset. Could the proposed correlation exist solely in a small subset of genes selected? To perform rigorous analysis of the dataset, the authors should randomize the selection of the genes and then rank them by HSF1 RNA level.

4. Fig 1 C, D, and E and Fig 2 B, C, and E all have 2 replicates. Additional replicates are needed to obtain statistical analysis.

Reviewer #2 (Comments to the Authors (Required)):

The authors have addressed my concerns with new data.

Reviewer #3 (Comments to the Authors (Required)):

This is the revised version of the manuscript by Silveira et al. where the authors presented evidence that low levels of stress response master regulator HSF1 is associated with breast cancer resistance. Reducing HSF1 re-establishes ER α levels along with the response to anti-estrogens. I appreciate the effort made by the authors that crystallizes in an increase in the number of Figures from 3 to 6 presented in the revised version of the manuscript. However, my concerns about this paper remain, the authors in my opinion have not addressed how HSF1 from a mechanistic point of view crosstalk with ER α and therefore the conceptual advance is limited. If the main objective of this study, as the authors indicate, is to present HSF1 inhibition as a potential avenue for treating endocrine resistant breast cancers, I consider that an exhaustive study on the involved mechanism and its experimental demonstration is required.

Minor points:

a) What happens to ER-repressed target genes when HSF1 levels change? Only estrogen-activated genes such as TFF1, GREB1, and PGR have been evaluated.

b) What concentration of the HSF1 inhibitor (KRIBB11) was used? Is the inhibitor present during the 5 days of the experiment? Did KRIBB11 affect ER levels? The details of how the experiment in Figure 6 was performed should be explained more clearly.

c) The size of the graphs in Figure 6 should be increased to help the visualization of the results.

Point-by-point response to reviewers

Reviewer #1:

The manuscript by Silveira et al. investigates the modulatory effect of heat shock factor 1 (HSF1) on estrogen receptor α (ER α)-dependent transcriptional program that leads to anti-estrogen resistance phenotype in breast cancer cells. In the revised version of the manuscript, the authors propose that overexpression of HSF1 triggers proteasomal degradation of ER α , negatively impacting ER α -dependent gene regulation. Further, the authors used HSF1 inhibitor KRIBB11 in addition to Fulv/TAM to show that anti-estrogen response is restored in ER α + endocrine-resistant breast cancers. However, some of the new experiments require clarification and additional experiments.

We thank the reviewers for the constructive comments. As requested, we clarified all raised issues and added the requested experiments.

1. One of the main conclusions of the study mechanistically is that overexpression of HSF1 leads to proteasomal degradation of ER α protein, thereby ablating its gene regulatory program. As evidence, the authors show that in response to HSF1 induction, ER α protein is degraded prior to cessation of proposed ER-mediated transcription. Further, in Fig 2E, ER α protein level is restored upon use of proteasome inhibitor MG-132. To connect the dots in a more than correlative fashion, the authors should show that treatment with proteasome inhibitor MG-132 restores ER α -dependent transcription program.

While an interesting idea, the cellular response to MG-132 is complex to address such a specific question. For example, treating MCF7 cells with MG-132 led to an increase in both HSF1 and ER α protein levels (Fig. 2G, lane 4). These results are in line with previous reports suggesting that proteasome inhibition is an important stress for the cell leading to activation of the HSF1-dependent stress program [Lee and Goldberg (PMID: 9418850); Kawazoe et al. 1998 (PMID: 9716376); Du et al. 2009 (PMID: 19006120); Kim et al. 2011 (PMID: 21738571)]. Nonetheless, we used the HSF1-overexpressing cells to test if the proteasome inhibitor MG-132 was sufficient to restore the ER α -dependent transcription program. Unfortunately, we could not establish that treating MCF7 cells with MG-132 in any condition led to an upregulation of ER α -activated genes. We believe that the broad effect of the MG-132 is too complex to study specific transcriptional response.

Along the same lines, to conclusively establish that HSF1 triggers proteasomal degradation of ER α , the authors should repeat the experiments with ER mutated at Lysine 302 and 303 (PMID: 18388150) which would be expected to be resistant.

To demonstrate a direct link between HSF1 overexpression and the proteasome-dependent degradation of ER α , we created a tagged (V5) version of the ER α K302, 303A mutant (Fig. 2H). Next, we generated stable MCF7 cell lines using the wild type and mutant ER α in which we overexpressed HSF1. The levels of exogenous HSF1 and ER α were measured using the V5 tag. The results showed that overexpression of HSF1 decreased levels of the wild type ER α , but not the K302,303A mutant. We could not determine why the co-transduction of HSF1 and the ER α mutant led to higher expression levels of both proteins. Nonetheless, the results demonstrate that the ER α mutant is resistant to degradation and support that HSF1 triggers proteasome-dependent degradation of ER α . The results were added to Figure 2H and the text was updated.

2. Further, it would be interesting to see how the dynamics of ER α binding at the respective genes change with HSF1 induction due to heat shock at different timepoints.

We are now showing ER α binding at the respective genes following heat shock. Levels of ER α were decreased between 50-70% after 30 minutes and 70-85% after 60 minutes. The results were added to Figure 2D and the text was updated.

The authors propose a regulatory mechanism of HSF1 negating ER α -dependent gene expression contrasting to the one proposed by Khalque et al 2008 (direct binding of HSF1 to ER α -target genes), these need to be addressed in detail.

Khalque et al 2008 proposed that HSF1 was directly repressing ER α target genes by recruiting the NuRD complex. In support of their conclusions, they presented evidence of chromatin recruitment of HSF1 and NuRD for a single gene, *TFF1* (pS2), following HSF1 and MTA1 overexpression in presence of heregulin. No quantification was provided (which is almost impossible to do accurately with the method used). A problem with this study is that overexpressing of a single component of a multisubunit complex can lead to unexpected consequences on the activity of the complex. Also, many recent reports do not support that model. Indeed, the NuRD complex is known to be binding virtually all active promoter and enhancer regions [de Dieuleveult et al. 2016 (PMID: 26814966); Gunther et al. 2013 (PMID: 23361464); Miller et al. 2016 (PMID: 27471257); Shimbo et al. 2013 (PMID: 24385926); Stevens et al. 2017 (PMID: 28289288) Wang et al. 2007 (PMID: 17392792); Kraushaar et al. 2018 (PMID: 30254051)]. For example, we have shown (Whyte et al. 2012 (PMID: 22297846)) that recruitment of the NuRD complex was necessary to decommission enhancers to modify cell state in embryonic stem cells. In MCF7 cells, the NuRD complex is known to occupy over 7,000 active promoter regions (Shimbo et al. 2013 (PMID: 24385926)) including *TFF1* and most ER α -activated genes (Response_to_reviewers Fig. 1). Therefore, HSF1-dependent recruitment of the NuRD complex at ER α -activated genes is a model not supported by current knowledge.

Nonetheless, we investigated whether HSF1 was recruited to ER α -activated genes after heat shock. At ER α -activated genes, HSF1 signal was 95-443-fold lower than canonical HSF1 targets, but slightly above background (Response_to_reviewers Fig. 2). Taken together, we do not have any evidence that HSF1 is responsible for the recruitment of the NuRD complex at ER α -activated genes. However, proving it without the shadow of a doubt would require a large amount of resources which would extend far beyond the scope of the manuscript.

Lastly, the reviewer is suggesting that our model contrast with the HSF1\NuRD model that was proposed. However, gene regulation often depends on the combination of mechanisms to activate and repress transcription. For example, repression at the promoter levels could preclude degradation of ER α . Furthermore, proteasome-dependent mechanisms have been well-known to recycle transcription factors involved in gene activation in addition to play other roles in transcriptional regulation (Geng et al. 2012 (PMID: 22404630)). Therefore, multiple levels of regulation are likely implicated in the control of the transcriptional program of breast cancer cells.

We added the requested details throughout the text to better describe the regulatory mechanism.

Also, does the induction of HSF1 in MCF7 cells upon heat shock lead to a similar transcriptional program as endocrine-resistant LCC9 cells or is there a difference in the regulatory mechanism? **While the reviewer is asking in appearance a simple question, the reality is a complicated answer. We refrained from using heat shock in the original submission because we are inducing a massive cellular stress response in which HSF1 is one of many activated**

transcription factors [Mahat et al. 2016 (PMID: 27052732); Duarte et al. 2016 (PMID: 27492368); Vihervaara et al. 2018 (PMID: 29556092)]. Furthermore, there are widespread effects on transcription, RNA stability and translation following heat shock that are HSF1-independent [Mahat et al. 2016 (PMID: 27052732); Vihervaara et al. 2018 (PMID: 29556092) and Vihervaara and Sistonen 2014 (PMID: 24421309)]. In addition, HSF1 targets in cancer cells and those following heat shock were shown to be different (Mendillo et al. 2012 (PMID: 22863008)). In fact, HSF1 interacting proteins are different in normal growth conditions compared to an acute stress (Burchfiel et al. 2020 (PMID: 33208463)). Therefore, trying to globally correlate the program of MCF7 cells stimulated with heat shock to LCC9 cells is impossible. However, HSF1-activated genes are induced and ER α -activated genes are decreased upon heat shock stimulation (Fig. 2B). These results agree with the general idea that activating HSF1 through heat shock leads to a decrease in the ER α -dependent program.

3. Fig S2. The authors should describe how subset of HSF1 (97) and ER α (42) signature genes were selected from a much larger pool of HSF1/ER α regulated genes from the dataset. Could the proposed correlation exist solely in a small subset of genes selected? To perform rigorous analysis of the dataset, the authors should randomize the selection of the genes and then rank them by HSF1 RNA level.

The subsets of HSF1 and ER α were selected as described in the Materials and Methods section using processed standardized expression datasets generated by the TCGA Research Network. Gene expression correlations between HSF1 and all other genes were calculated using the Pearson correlation function (cor under R). Genes with correlations above 0.3 and below -0.3 were selected for the representation in Figure S2A. Therefore, the correlation with HSF1 RNA levels is already taken into account in the analysis. We updated the Materials and Methods and Figure legend to clearly describe how the genes were selected.

To directly address whether the correlation exists because we selected a subset of genes, we lowered the correlation threshold to increase the number of genes regulated by HSF1 and ER α (Fig. S2B). With the |0.3| cutoff (total: 96 genes, Fig S2), 87% of HSF1-regulated genes were correlated with HSF1 RNA levels. When the threshold was lowered at |0.2| (total: 231 genes) and |0.1| (total: 439 genes), this number was reduced to 75% and 67% respectively for HSF1-regulated genes. For ER α -regulated genes, 81% were anticorrelated with HSF1 RNA levels for a |0.3| threshold. The percentage was 73% and 67% at the |0.2| and |0.1| thresholds, respectively. Therefore, even by significantly increasing the subset of genes, the conclusion is the same: HSF1-regulated genes are correlated with HSF1 levels while ER α -regulated genes are anti-correlated. The results were added to Figure S2B.

4. Fig 1 C, D, and E and Fig 2 B, C, and E all have 2 replicates. Additional replicates are needed to obtain statistical analysis.

We added a 3rd replicates and updated the statistical analysis.

Reviewer #2:

The authors have addressed my concerns with new data.

We thank reviewer #2 for the constructive criticism during the review process.

Reviewer #3:

This is the revised version of the manuscript by Silveira et al. where the authors presented evidence that low levels of stress response master regulator HSF1 is associated with breast cancer resistance. Reducing HSF1 re-establishes ER α levels along with the response to anti-estrogens.

I appreciate the effort made by the authors that crystallizes in an increase in the number of Figures from 3 to 6 presented in the revised version of the manuscript. However, my concerns about this paper remain, the authors in my opinion have not addressed how HSF1 from a mechanistic point of view crosstalk with ER α and therefore the conceptual advance is limited. If the main objective of this study, as the authors indicate, is to present HSF1 inhibition as a potential avenue for treating endocrine resistant breast cancers, I consider that an exhaustive study on the involved mechanism and its experimental demonstration is required.

We thank the reviewer for providing a final round of comments. We are now describing additional conceptual advances (see point-by-point response to Reviewer #1). Therefore, we believe that we overcome all issues. Minor points are addressed below.

Minor points:

a) What happens to ER-repressed target genes when HSF1 levels change? Only estrogen-activated genes such as TFF1, GREB1, and PGR have been evaluated.

This issue was not raised in the previous round of review. However, we tested three well known ER α -repressed genes (*SMAD6*, *CCNG2* and *MMD*) (Stossi et al. 2009 (PMID:19188451)). Following estrogen stimulation, these genes were shown to be rapidly bound by ER α and repressed (Stossi et al. 2009 (PMID:19188451)). We measured expression of these genes in HSF1-overexpressing MCF7. Interestingly, the overexpression of HSF1 was associated with increased expression of *SMAD6*, *CCNG2* and *MMD*. These results suggest that overexpressing HSF1 decreased ER α levels which released the repression. We added the results to Figure 1B and updated the text.

b) What concentration of the HSF1 inhibitor (KRIBB11) was used?

The KRIBB11 concentration (1 μ M) was added to the Figures 6 and S4 legend.

Is the inhibitor present during the 5 days of the experiment?

Yes. Cells were cultured with the combination of KRIBB11 (1 μ M) and antiestrogens for the indicated number of days. We updated the figure legend to make the point clear.

Did KRIBB11 affect ER levels?

Low concentrations of KRIBB11 (lower than 5 μ M) decreased the HSF1 transcriptional activity while high concentrations (above 10 μ M) reduced HSF1 levels in multiple cellular models [Yoon et al. 2014 (PMID: 21078672); Mani et al. 2015 (PMID: 26100943); Samarasinghe et al. 2014 (PMID: 24291777); Antonietti et al. 2017 (PMID: 27777286); Fok et al. 2018 (PMID: 29391353); Kang et al. 2019 (PMID: 31540279); Carpenter et al. 2017 (PMID: 29088759)]. In MCF7 cells, 1 μ M was sufficient to reduce HSF1 levels which was associated with lower ER α levels (Fig. 6A and Response_to_reviewers Fig. 3A). In LCC9 cells, 10 μ M of KRIBB11 was required to decrease HSF1 levels (Response_to_reviewers Fig. 3B). At 1 μ M, levels of HSF1 were not affected and ER α was slightly reduced (Fig. 6C and Response_to_reviewers Fig. 3B). These results suggest that inhibition of HSF1 restores antiestrogen response by mechanisms different from HSF1 depletion (Fig. 4 and 5). We are speculating that HSF1 is destabilizing ER α independently from its transcriptional activity.

Therefore, HSF1 depletion restores ER α expression levels while inhibition does not. Why inhibition of HSF1 is not increasing ER α levels, but still restoring response to antiestrogen in LCC9 cells is an open question. Additional experiments will be required to specifically address this question in the future. The text was updated to highlight the difference between HSF1 depletion and inhibition in the context of restoring the antiestrogen response.

The details of how the experiment in Figure 6 was performed should be explained more clearly. We thank the reviewer for picking up the missing details. We updated the figure legend and the Materials and Methods section to clearly explain how the experiment was performed.

c) The size of the graphs in Figure 6 should be increased to help the visualization of the results. We improved the readability of Figure 6.

Figure 1. The NuRD complex is recruited to the regulatory regions of ER α -activated genes in MCF7 cells. Multiple subunits of the NuRD complex (MTA1, MTA2, MTA3 and MBD2) are recruited to the regulatory regions of ER α -activated genes, including *ESR1*, *TFF1* and *GREB1*, in standard MCF7 growth conditions. Enriched regions for the NuRD subunits were downloaded from the ENCODE project and visualized using the UCSC Genome Browser. Below the gene depictions, enriched regions for each NuRD subunit are represented by gray boxes.

Figure 2. HSF1 is not recruited to the regulatory regions of ER α -activated genes following heat shock in MCF7 cells. While high levels of HSF1 are detected at HSF1-activated genes (*HSPH1*, *HSP90AA1* and *HSPD1*) following heat shock, background levels are found at ER α -activated genes (*ESR1*, *TFF1* and *GREB1*). *Left* - ChIP-qPCR results from Figure 2C showing HSF1 recruitment to HSF1-activated genes. *Right* - Magnification of the results presented in Figure 2C showing an HSF1 signal close to background level at ER α -activated genes. Percentage of input means \pm SEM ($n = 3-5$ biological replicates) are represented and P -values were calculated using bivariate t-test. (*) $P \leq 0.05$.

Figure 3. Concentration-dependent effects of KRIBB11 on HSF1 and ER α expression levels. (A) MCF7 and (B) LCC9 cells were treated with 1 μM and 10 μM of the HSF1 inhibitor KRIBB11 for 24 h prior to total protein extraction. Protein quantifications were performed using the ImageJ software and GAPDH as loading control. NA: No available quantification.

January 27, 2021

RE: Life Science Alliance Manuscript #LSA-2020-00811-TRR

Dr. Steve Bilodeau
Centre de Recherche du CHU de Québec - Université Laval
9 McMahon Street
Québec, Quebec G1R2J6
Canada

Dear Dr. Bilodeau,

Thank you for submitting your revised manuscript entitled "Modulating HSF1 levels impacts expression of the estrogen receptor α and antiestrogen response". We would be happy to publish your paper in Life Science Alliance pending final revisions necessary to meet our formatting guidelines.

Along with the points listed below, please also attend to the following:

- please consult our manuscript preparation guidelines <https://www.life-science-alliance.org/manuscript-prep> and make sure your manuscript sections are in the correct order
- please separate Results and Discussion into 2 separate sections - 1. Results and 2. Discussion
- please add a conflict of interest statement to your main manuscript text
- please add a callouts for Figure S2 A,B to your main manuscript text
- please upload your main manuscript text as an editable doc file
- please use the [10 author names, et al.] format in your references (i.e. limit the author names to the first 10)
- please provide the original unedited source data for HSF1 blots shown in Figure 1A and S1A, and ER α blot in Figure S3A

A. FINAL FILES:

-- High-resolution figure, supplementary figure and video files uploaded as individual files: See our

detailed guidelines for preparing your production-ready images, <https://www.life-science-alliance.org/authors>

B. MANUSCRIPT ORGANIZATION AND FORMATTING:

Sincerely,

Shachi Bhatt, Ph.D.
Executive Editor
Life Science Alliance
<https://www.lsjournal.org/>
Tweet @SciBhatt @LSAJournal

Reviewer #1 (Comments to the Authors (Required)):

The authors have made a significant effort to address the comments of this Reviewer and provide new data in the revision that strengthens the study considerably. I have no further comments.

Reviewer #3 (Comments to the Authors (Required)):

The authors have addressed my concerns by adding new data and clarifying the methods used in the experiments with the HSF1 inhibitor.

February 3, 2021

RE: Life Science Alliance Manuscript #LSA-2020-00811-TRRR

Dr. Steve Bilodeau
Centre de Recherche du CHU de Québec - Université Laval
9 McMahon Street
Québec, Quebec G1R2J6
Canada

Dear Dr. Bilodeau,

Thank you for submitting your Research Article entitled "Modulating HSF1 levels impacts expression of the estrogen receptor α and antiestrogen response". It is a pleasure to let you know that your manuscript is now accepted for publication in Life Science Alliance. Congratulations on this interesting work.

DISTRIBUTION OF MATERIALS:

Again, congratulations on a very nice paper. I hope you found the review process to be constructive and are pleased with how the manuscript was handled editorially. We look forward to future exciting submissions from your lab.

Sincerely,

Shachi Bhatt, Ph.D.

Executive Editor

Life Science Alliance

<https://www.lsjournal.org/>

Interested in an editorial career? EMBO Solutions is hiring a Scientific Editor to join the international Life Science Alliance team. Find out more here -

https://www.embo.org/documents/jobs/Vacancy_Notice_Scientific_editor_LSA.pdf